# DistillSpec: Improving Speculative Decoding via Knowledge Distillation

**Yongchao Zhou**[1,3]*, **Kaifeng Lyu**[1,4]*, **Ankit Singh Rawat**[1], **Aditya Krishna Menon**[1],
**Afshin Rostamizadeh**[1], **Sanjiv Kumar**[1], **Jean-François Kagy**[1]†, **Rishabh Agarwal**[2,5]†
[1]Google Research   [2]Google DeepMind   [3]University of Toronto   [4]Princeton University   [5]Mila

## Abstract

Speculative decoding (SD) accelerates large language model inference by employing a faster *draft* model for generating multiple tokens, which are then verified in parallel by the larger *target* model, resulting in the text generated according to the target model distribution. However, identifying a compact draft model that is well-aligned with the target model is challenging. To tackle this issue, we propose *DistillSpec*, a method that uses knowledge distillation to better align the draft model with the target model before applying SD. DistillSpec makes two key design choices, which we demonstrate via systematic study to be crucial to improving the draft and target alignment: utilizing *on-policy* data generation from the draft model, and *tailoring the divergence function* to the task and decoding strategy. Notably, DistillSpec yields $10 - 45\%$ speedups over standard SD on a range of benchmarks, using both greedy and non-greedy sampling. We show that the distilled model can be well transferred to various tasks with an average speedup of $26\%$. Furthermore, we combine DistillSpec with lossy SD to achieve fine-grained control over the latency vs. task performance trade-off. Finally, in practical scenarios with models of varying sizes, first using distillation to boost the performance of the target model and then applying DistillSpec to train a well-aligned draft model can reduce decoding latency by $6 - 10\times$ with minimal performance drop, compared to standard decoding without distillation.

## 1 Introduction

Large language models (LLMs) have revolutionized natural language understanding and generation across diverse applications (OpenAI, 2023; Anil et al., 2023). However, their autoregressive generation nature poses significant computational challenges, especially in real-time deployments with stringent latency constraints (Thoppilan et al., 2022; Pope et al., 2023). Conversely, smaller language models, while computationally efficient, often lack the expressive power of their larger counterparts and achieve subpar performance. While reducing the inference cost of larger models, e.g., via quantization or pruning, or improving the performance of the smaller models, e.g., via knowledge distillation (KD) (Hinton et al., 2015), constitute natural approaches to enable a favorable performance versus inference cost trade-off, these approaches frequently result in unacceptable performance gap compared to the high-quality large models. This has inspired a growing literature on designing mechanisms that combine both large and small models at inference to approximate the performance of the larger models without incurring their high computational cost.

Among conventional approaches, model cascading aims to identify easy instances where smaller models suffice to achieve good performance, and soliciting larger models only on a subset of hard instances (Rowley et al., 1998; Xu et al., 2014) or tasks (Cai et al., 2023b). Different from such task- or instance-level cascading, *speculative decoding* (SD) (Leviathan et al., 2023; Chen et al., 2023) exploits the token-level variability in the computation demand during LLM inference by interactively invoking a small "draft" model and a large "target" model. At a given stage during inference, the draft model generates successive candidate tokens for multiple inference steps via autoregressive decoding. The target model then verifies the candidate tokens via parallel decoding, and employs rejection sampling to accept a subset of candidate tokens at contiguous positions.

The main objective of SD is to speed up text generation while guaranteeing that the decoded tokens follow the target model distribution. SD relies on the insight that the combined cost of autoregressive decoding with a small draft model followed by parallel verification with the target model is

---

*Student Researcher at Google Research.   †Advising contribution.   Corresponding authors: <yczhou@cs.toronto.edu>, <jfkagy@google.com>, and <rishabhagarwal@google.com>.

Figure 1: Performance comparison of standard speculative decoding (SD) vs. our proposed DistillSpec, with small- and XL-sized models from the T5 v1.1 family (Raffel et al., 2020) being utilized as the draft and the target models, respectively. DistillSpec enhances SD speed by better aligning the draft with the target via white-box knowledge distillation, resulting in a consistent $10-45\%$ speedup improvement over standard SD across various datasets. The distilled draft model from GSM8K transfers well to 23 unseen BigBenchHard tasks (Suzgun et al., 2022), resulting in an average speedup of $26\%$. See § 5.1 for additional details.

lower than the cost of autoregressive decoding with the target model alone. However, the realized inference cost reduction or latency improvement crucially depends on the *acceptance rate* of the draft-generated tokens by the target model, which can be shown to be directly tied to the alignment between the token distributions of the draft and target models. Thus, a successful application of SD hinges on identifying a compact draft model that simultaneously has small autoregressive decoding cost *and* is closely aligned with the target model.

In this work, we propose DistillSpec, a novel approach that relies on KD (Hinton et al., 2015) to obtain an effective draft model. Unlike the standard application of KD which primarily focuses on improving the task performance of a small student model, DistillSpec aims at aligning the student (draft) model with the teacher (target) model to enhance the acceptance rate during SD. We undertake a comprehensive exploration of the distillation process for speeding up SD, considering several factors including the composition of training data, choice of divergence functions to define the training objective for KD, and decoding strategies. Notably, our findings underscore that using model-generated data is crucial for ensuring strong student-teacher alignment across various tasks via KD, and that the selection of the best-performing divergence function in DistillSpec is highly task-dependent and sensitive to the decoding strategy (i.e., greedy versus non-greedy). Furthermore, we explore the utility of DistillSpec for lossy SD (Leviathan et al., 2023) which allows for sampling away from the target model distribution. We show that combining DistillSpec with lossy SD enables a more fine-grained control over the latency versus task performance trade-off.

Finally, we carry out a systematic study of how to design an efficient inference scheme in a practical setting where one has access to multiple models of increasing size and quality. Leveraging the insights that we have laid out about KD and SD, our study concludes that the most effective strategy involves first distilling a large model into a smaller one as the potential target model for performance optimization, followed by DistillSpec for distilling an even smaller model to be used as the draft model in SD. This approach results in a remarkable $6-10\times$ reduction in latency, compared to a standalone non-distilled target model of the same size, with minimal performance degradation.

Our key contributions are:

(i) We propose DistillSpec, a method that uses KD to enhance draft model alignment with the target model (§4), and show that our method can improve SD speed by $10-45\%$ while preserving model performance across diverse datasets under greedy and non-greedy sampling (Figure 1).

(ii) We conduct an extensive analysis of the optimal distillation recipe (§5.2) for model alignment, encompassing factors such as training data generation and different divergences, and emphasizing the distinctions between standard KD and distillation tailored for SD.

(iii) We extend DistillSpec to lossy SD, enabling refined control over the quality-latency trade-off. Moreover, we offer insights for combining KD and SD when several models are available (§5.3).

## 2 RELATED WORK

**Speculative decoding (SD).** Due to the inherent sequential nature of autoregressive decoding, the primary latency bottleneck in LLM inference arises from memory read/write operations rather than arithmetic computations (Pope et al., 2023). Speculative decoding (Leviathan et al., 2023;

Chen et al., 2023) (SD) addresses this challenge by utilizing a compact draft model to generate a batch of tokens sequentially, while validating them in parallel with a larger target model. Prior to SD, various parallel computing paradigms have been explored for autoregressive models, including block parallel sampling (Stern et al., 2018), shallow aggressive decoding (Sun et al., 2021), and aggressive decoding (Ge et al., 2022). However, these approaches are not readily adaptable to typical language models due to potential deviations from target model's distribution, strict input constraints, or limited support for general stochastic sampling. Notably, recent variants of SD have considered different interactions between the draft and target model to reduce unnecessary computation (Kim et al., 2023) and incorporated parallel computation along the batch axis, sometimes combined with token tree verification, as seen in SpecTr (Sun et al., 2023), SpecInfer (Miao et al., 2023), and Medusa (Cai et al., 2023a). In contrast, our work focuses on enhancing SD by improving the alignment between the small draft model and the large target model through KD, which does not require any changes to serving infrastructures already implementing SD and is complementary to the recent variants of SD. Furthermore, we conduct a systematic study of lossy SD for providing nuanced control over the trade-off between quality and latency for specific serving models.

**Knowledge distillation (KD) for LLMs.** KD (Buciluă et al., 2006; Hinton et al., 2015), which trains high-quality smaller student models with the supervision of larger teacher models, has emerged as a vital technique for reducing inference cost while maintaining model quality across a range of domains. In the context of LLMs, prior uses of KD (Taori et al., 2023; Fu et al., 2023) have mostly focused on *black-box* KD, wherein only the teacher's output generations, generally via APIs, are accessible during student training. However, with the proliferation of open-source LLMs (Zhang et al., 2022; Touvron et al., 2023), which enable access to teacher weights and logits, there is a growing interest in *white-box* KD. White-box KD allows student models to benefit from richer supervision signals provided by white-box teacher models, leading to enhanced language abilities (Agarwal et al., 2023; Gu et al., 2023; Wen et al., 2023).

Unlike prior works focused on creating highly capable standalone student models, we harness KD to foster closer collaboration between smaller and larger models in SD, which may be particularly valuable when a small distilled model alone cannot meet stringent quality requirements. While Stern et al. (2018) use a black-box KD approach (SeqKD) to enhance blockwise parallel decoding, their samples are generated from the large target model, which is prohibitively expensive for LLMs. Furthermore, they ignore the teacher model's logits and train their draft model using only one-hot teacher labels—a reasonable choice for greedy decoding but a less effective one for non-greedy sampling (Figure 2). Concurrently, Liu et al. (2023) propose to improve SD using KD, but they assume an online setup with a changing query distribution, and focus on improving the acceptance rate rather than reducing the actual latency.

## 3 BACKGROUND: SPECULATIVE DECODING

**Notation.** Given an input sequence $x$ comprising tokens from a pre-defined vocabulary, a language model $\mathcal{M}$ provides a distribution over possible output sequences $y$. Suppose we employ SD with a **compact draft model** $\mathcal{M}_q$ to assist a **larger target model** $\mathcal{M}_p$. Let $p(y_t \mid x, y_{<t})$ and $q(y_t \mid x, y_{<t})$ represent the distributions governing next-token predictions at time step $t$ for $\mathcal{M}_p$ and $\mathcal{M}_q$, respectively, given the context $\rho = \{x, y_{<t}\}$. Given input $x$ as prefix, let $p_{\leq T}(y \mid x)$ and $q_{\leq T}(y \mid x)$ represent the distributions governing the sequence $y$ sampled autoregressively from $\mathcal{M}_p$ and $\mathcal{M}_q$, respectively, where the generation stops either when an end-of-sequence token is sampled or the maximum sequence length $T$ is reached. For simplicity, we use $p(y_t)$ and $q(y_t)$ as shorthands for $p(y_t \mid x, y_{<t})$ and $q(y_t \mid x, y_{<t})$, whenever the context $\rho$ is clear. Similarly, $p_{\leq T}(y)$ and $q_{\leq T}(y)$ serve as shorthands for $p_{\leq T}(y \mid x)$ and $q_{\leq T}(y \mid x)$, whenever the input $x$ is clear.

**Speculative sampling.** Standard SD uses a procedure called *speculative sampling* to generate tokens from the draft model while maintaining the same output distribution as the target model. As detailed in Algorithm A.1 (Appendix), each step of SD works as follows. First, a *block* of $\gamma$ tokens, denoted as $y_t, \ldots, y_{t+\gamma-1}$, is autoregressively sampled from $q(y_t), \ldots, q(y_{t+\gamma-1})$. Next, the $\gamma$ tokens are verified in parallel by passing them to $\mathcal{M}_p$ as a whole block, which sequentially accepts token $y_{t+i}$ with probability $\min\left(1, p(y_{t+i})/q(y_{t+i})\right)$. If any token $y_{t+i}$ is rejected before the end of the block, the subsequent tokens are discarded and the rejected token is resampled from the adjusted distribution $p'(y_{t+i}) \propto \max(0, p(y_{t+i}) - q(y_{t+i}))$; otherwise, the drafted tokens are all accepted and an extra token is sampled from $p(y_{t+\gamma})$ and appended to the output sequence. This process guarantees that the sequence of accepted and resampled tokens follow the same output distribution

as $p(y_{t+i})$ (Leviathan et al., 2023). The procedure is repeated until an end-of-sequence token is accepted, or the maximum sequence length $T$ has been reached.

**Efficiency measure: acceptance rate.** Each SD step takes a constant amount of time, so the wall-clock time scales linearly with the number of steps. This number is equal to the total number of times that the target model rejects a token, plus the number of blocks accepted as a whole, where the latter term is small for large $\gamma$. This motivates us to use the *acceptance rate* as a surrogate efficiency measure for the wall-clock time. For an ideal SD process with $\gamma = \infty$, we define the **sequence-level acceptance rate** $\alpha(x)$ for a given input $x$ as follows:

$$\alpha(x) := \frac{\mathbb{E}\left[\text{number of accepted tokens in generating } y\right]}{\mathbb{E}\left[\text{number of tokens in } y\right]} = \frac{\mathbb{E}_{y \sim p_{\leq T}(y \mid x)}\left[\sum_{t=1}^{|y|} \beta(x, y_{<t})\right]}{L_p(x)}, \quad (1)$$

where $\beta(x, y_{<t}) := \mathbb{E}_{y_t \sim q(y_t)}\left[\min\left(1, p(y_t)/q(y_t)\right)\right]$ is the token-level acceptance rate, and expectations are taken over the randomness in SD. Since speculative sampling preserves the output distribution of the target model, the denominator is simply equal to the expected length of the target output $L_p(x) := \mathbb{E}_{y \sim p_{\leq T}(y \mid x)}\left[|y|\right]$, which is invariant to the choice of draft model. Therefore, the acceptance rate is directly related to the expected total number of rejected tokens $(1 - \alpha(x)) \cdot L_p(x)$, which lower bounds the expected number of SD steps.

**Efficiency measure: block efficiency.** In practice, SD is usually employed with a fixed finite block size $\gamma$; thus, a more relevant efficiency measure is the **block efficiency** $\tau$. Given an input $x$, we compute the block efficiency $\tau(x)$ as the expected number of accepted tokens per block. Note that, for a given block size $\gamma$, the maximum value of $\tau(x)$ is $\gamma + 1$, corresponding to the case where all drafted tokens are accepted and augmented with an additional token sampled by the target model. If we assume that the token-level rates $\beta(x, y_{<t})$ are i.i.d., then the sequence-level acceptance rate satisfies $\alpha = \mathbb{E}[\beta]$ and $\tau(x) = (1 - \alpha^{\gamma+1})/(1 - \alpha)$ (Leviathan et al., 2023).

**Wall-clock time improvement**. For given block efficiency $\tau(x)$, the expected speedup of SD is given by $\tau(x)/(c\gamma + 1)$, where the relative latency $c$ is the ratio between the times elapsed when making a single forward pass through the draft model $\mathcal{M}_q$ and through the target model $\mathcal{M}_p$.

# 4 DISTILLSPEC: KNOWLEDGE DISTILLATION FOR SPECULATIVE DECODING

As described in § 3, speculative decoding (SD) leverages a small (draft) model to reduce the latency of decoding from the larger (target) model distribution without any performance drop. However, the realized speedup critically depends on how "well-aligned" the draft model is to the target model. In this work, we propose **DistillSpec**, a general framework that improves SD by better aligning the target model and draft model using white-box knowledge distillation (KD). We first present KD-based training of the draft model, and highlight how our objective of enhancing SD via KD influences our selection of training data generation method and divergence function—two key ingredients of DistillSpec. We then discuss how DistillSpec can be extended to lossy SD.

Let the draft model $\mathcal{M}_q^\theta$ be parameterized by $\theta$. DistillSpec utilizes predictions from the target model $\mathcal{M}_p$ as a source of supervision in training the draft model $\mathcal{M}_q^\theta$. We assume white-box access to both models, i.e., we can obtain their next-token distributions $p(y_t)$ and $q(y_t)$, and therefore we are able to generate samples from both models. Given a divergence function $D$ that measures the misalignment between two distributions, KD-based training of the draft model seeks to minimize the divergence between the teacher (target) and student (draft) distributions over a training set $\mathcal{G}$:

$$\theta^* := \arg\min \mathbb{E}_{(x,y) \sim \mathcal{G}}\left[D(\mathcal{M}_p \| \mathcal{M}_q^\theta)(y|x)\right], \quad (2)$$

where $D(\mathcal{M}_p \| \mathcal{M}_q^\theta)(y|x) = \frac{1}{|y|} \sum_{t=1}^{|y|} D\left(p(\cdot \mid x, y_{<t}) \| q^\theta(\cdot \mid x, y_{<t})\right)$. We note that flexibility in how $\mathcal{G}$ is constructed and the choice of divergence $D$ gives rise to different possible KD algorithms. For instance, $\mathcal{G}$ may consist of task-specific input-output pairs $(X, Y)$ or sequences generated from $\mathcal{M}_p$ or $\mathcal{M}_q^\theta$. While *forward* KL ($D_{\text{FKL}}$) is the commonly used divergence for KD (Hinton et al., 2015), recent works (Agarwal et al., 2023; Wei et al., 2022) have shown the effectiveness of alternative divergences, including *reverse* KL ($D_{\text{RKL}}$), Jensen–Shannon divergence ($D_{\text{JSD}[\beta]}$), and total variation distance ($D_{\text{TVD}}$). Further details on each divergence can be found in Appendix A.1. Table 1 summarizes various distillation recipes, each being a specialized instance of Algorithm A.2.

Table 1: Summary of various KD algorithms in terms of training data $\mathcal{G}$ and divergence $D$ (cf. Eq. 2). Wen et al. (2023); Agarwal et al. (2023) also consider other $D$; we list the most representative one.

| Name | Divergence | Training Data ($\mathcal{G}$) |
|---|---|---|
| SeqKD (Kim & Rush, 2016) (**Black-box KD**) | FKL | Data generated by teacher $\mathcal{M}_p$ |
| Supervised KD (Hinton et al., 2015) | FKL | Fixed dataset of input-output pairs |
| ImitKD (Lin et al., 2020) | FKL | Fixed dataset + data generated by $\mathcal{M}_q^\theta$ |
| f-Distill (Wen et al., 2023) | TVD | Data generated by $\mathcal{M}_q^\theta$ and $\mathcal{M}_p$ |
| On-policy GKD (Agarwal et al., 2023) | FKL / JSD | On-policy data from student $\mathcal{M}_q^\theta$ |

Our choices for $\mathcal{G}$ and $D$ are guided by how the resulting distilled model, once employed as draft model, improves the speed of SD. Towards this, we first highlight the role that $D_{\text{TVD}}$ between $p(y_t)$ and $q(y_t)$ plays in dictating the acceptance rate (§ 3)—a key efficiency measure for SD.

**TVD as proxy for the acceptance rate.** Leviathan et al. (2023, Corollary 3.6) show that the token-level acceptance rate $\beta(x, y_{<t})$ satisfies $\beta(x, y_{<t}) = 1 - D_{\text{TVD}}(p(y_t), q(y_t))$. Hence, Eq. 1 implies that maximizing the sequence-level acceptance rate $\alpha(x)$ is equivalent to minimizing the expected $D_{\text{TVD}}$ between $p(y_t)$ and $q(y_t)$ over the output sequence distribution of $\mathcal{M}_p$, i.e.:

$$\alpha(x) = 1 - \mathbb{E}_{y \sim p_{\leq T}(y \,|\, x)} \left[ \sum_{t=1}^{|y|} D_{\text{TVD}}(p(y_t), q(y_t)) \right] / L_p(x). \tag{3}$$

**Choice of divergence.** Based on Eq. 3, it appears that directly minimizing $D_{\text{TVD}}$ may be a principled objective for draft model distillation. While optimizing $D_{\text{TVD}}(p, q)$ is theoretically inspired, our empirical study shows that such an objective may not consistently yield optimal results. We find that the choice of the most suitable divergence is highly task-dependent (§ 5.2).

**Choice of training data.** As for $\mathcal{G}$, one could resort to an existing ground-truth dataset, however the teacher's output distribution may deviate from the ground-truth distribution despite the teacher having been fine-tuned on it. Moreover, ground-truth datasets are often limited in size, so training *only* on such data could result in overfitting. To mitigate these issues, we use model-generated outputs for distillation. Specifically, we prompt the model with a task input sampled from a ground-truth training dataset, and use the model response as data for distillation. Both the teacher and student models may be used to generate the distillation examples.

**Model-generated distillation data.** Eq. 3 suggests optimizing the expected $D_{\text{TVD}}$ over outputs generated from the teacher. Decoding from a large teacher is generally prohibitively expensive, especially at the scale of dataset required for KD. To reduce the generation cost, we explore using *on-policy data* during distillation, i.e., output sequences sampled from the student itself. Besides being more computationally efficient compared to teacher generations, this approach is motivated by Gu et al. (2023); Agarwal et al. (2023), who have shown that distilling on on-policy data improves student task performance. However, different from these prior works, our primary focus is on improving the student-teacher alignment. Thus, it may not be immediately clear whether minimizing the expected $D_{\text{TVD}}$ over on-policy (student-generated) data ensures an improved acceptance rate, which is computed as an expectation over the *teacher's* output distribution (cf. Eq. 3). Our following result shows that this is indeed the case.

**Theorem 4.1.** *For SD, if the draft model $\mathcal{M}_q^\theta$ achieves on-policy KD loss $\epsilon = \mathbb{E}_{x \sim X, y \sim q_{\leq T}(y \,|\, x)} \left[ D_{\text{TVD}}(\mathcal{M}_p \| \mathcal{M}_q^\theta)(y \,|\, x) \right]$, then the sequence-level acceptance rate is at least*

$$\mathbb{E}_{x \sim X} \left[ \alpha(x) \right] \geq 1 - T \cdot \mathbb{E}_{x \sim X} \left[ \frac{T}{L_p(x)} \right] \epsilon. \tag{4}$$

*When the target output length is always $T$, the bound simplifies to $\mathbb{E}_{x \sim X} \left[ \alpha(x) \right] \geq 1 - T\epsilon$.*

We defer the proof to Appendix A.2. Intuitively, it builds upon the following insights. If the on-policy KD loss is small, then, for any $1 \leq t \leq T$, the same loss evaluated only at the $t$-th token should also be small. Since the first token generation is independent of any other tokens, a small value of on-policy KD loss ensures that the first token distributions of the draft and target models are close. Then, an inductive argument shows that once the draft and target are similar on the first $t$ tokens, the distributions of the $(t+1)$-th token should also be close. Our proof makes this argument rigorous by utilizing variational representations of $D_{\text{TVD}}$, leading to a linear error bound in $T$.

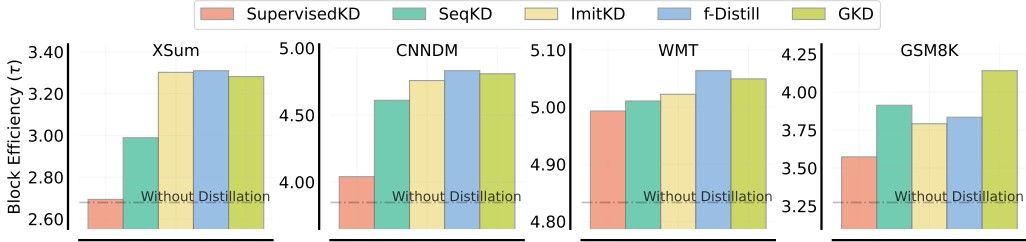

Figure 2: Distillation enhances block efficiency ($\tau$) across diverse datasets, highlighting the superiority of model-generated data over fixed ground truth data (SupervisedKD) and emphasizing the importance of white-box distillation in addressing SeqKD's subpar performance.

**DistillSpec with lossy SD.** DistillSpec enhances SD efficiency without any quality loss compared to the target model $\mathcal{M}_p$. In practical applications, a reduction in model quality may be justified in order to support even faster inference. For such scenarios, we extend DistillSpec to lossy SD (Leviathan et al., 2023), which uses a *lenience function* $f(p, \epsilon)$ that modifies the acceptance probability from $\min(1, p(y_t)/q(y_t))$ to $\min(1, f(p(y_t), \epsilon)/q(y_t))$ (cf. § 3). Here $f : [0, 1]^2 \to \mathbb{R}^+$ is increasing and decreasing in its first and second arguments, respectively, and $\epsilon \in [0, 1]$ is a free parameter (cf. Algorithm A.1). In this work, we evaluate multiple choices for the lenience functions: $f_{\text{lin}}(p, \epsilon) = p/\epsilon$, $f_{\text{sq}}(p, \epsilon) = p/\epsilon^2$, and $f_{\text{exp}}(p, \epsilon) = p^\epsilon$. For example, when the lenience function is $f_{\text{sq}}(p, \epsilon)$ and $\epsilon = 0.1$, token $x$ sampled from $q(y_t)$ becomes hundred times more likely to be accepted by the target, thus enabling faster inference at the expense of a potential drop in generation quality. Lenience was discussed by Leviathan et al. (2023) in the context of $f_{\text{lin}}$ and their treatment focuses solely on latency improvements, whereas we explore the use of different lenience functions as a precise control mechanism to achieve the desired quality-latency profile.

## 5 EXPERIMENTS

### 5.1 ENHANCING SPECULATIVE DECODING THROUGH DISTILLATION

We evaluate the effectiveness of KD in improving the speed of speculative decoding (SD). We follow Leviathan et al. (2023) and investigate its impact on the acceptance rate $\alpha$, block efficiency $\tau$, and actual latency speedup with a batch size of 1 under greedy sampling ($T = 0$) and standard temperature sampling ($T = 1$).

**Tasks and models.** Following Leviathan et al. (2023), we evaluate two model types: 1) GPT-like decoder-only Transformer models trained on LM1B task (Chelba et al., 2013) using a standard autoregressive objective, where the target and draft models have 234M and 33M parameters, respectively; and 2) standard encoder-decoder T5 v1.1 models (Raffel et al., 2020) fine-tuned on four different tasks, with T5-XL (3B) and T5-Small (77M) serving as the target and draft models, respectively. We utilize two datasets from the T5 paper, namely WMT EnDe (Bojar et al., 2014) and CNN/DM (Hermann et al., 2015), which deal with translation and text summarization, respectively. The remaining two tasks used to test T5 models are XSum (Narayan et al., 2018) and GSM8K (Cobbe et al., 2021), which deal with abstractive summarization and arithmetic reasoning, respectively. See Appendix B for more details.

**Decoding speedup.** Figure 1 shows that the impact of distillation on SD speed is evident, consistently yielding a $10-46\%$ improvement across various datasets. This effect is most pronounced when employing greedy decoding. The performance of KD for different block sizes and decoding strategies across five datasets is presented in Table C.1 (Appendix). These findings demonstrate that KD significantly enhances the acceptance rate and block efficiency of SD for both decoder-only and encoder-decoder models across all datasets. Distillation algorithms utilizing model-generated data consistently outperform other approaches, resulting in $\sim 20\%$ additional speedup compared to standard SD on LM1B, XSum, CNN/DM, and GSM8K.

**Block efficiency.** Figure 2 presents a comparison of block efficiency across different algorithms, employing temperature sampling ($T = 1$) with a block size $\gamma = 7$. The figure underscores the utility of model-generated data: using ground-truth data (i.e., Supervised KD) ranks lowest across all settings. In contrast, $f$-Distill and GKD, which only use model-generated data, significantly outperform other KD variants. The subpar performance of SeqKD, despite being purely trained on data

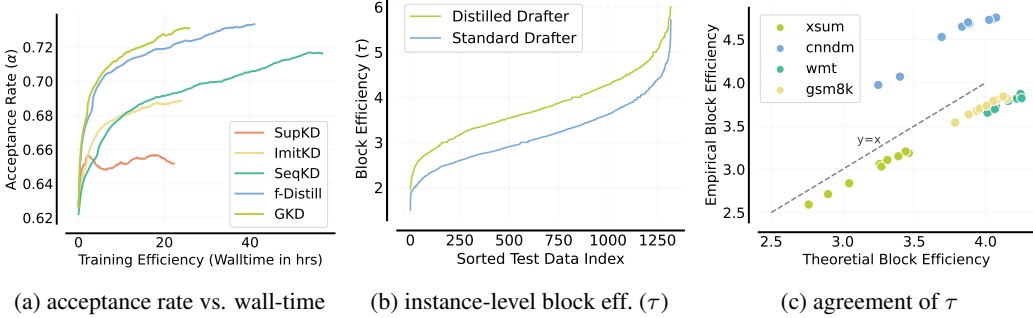

Figure 3: (a) White-box KD using teacher logits and model-generated data is crucial. Draft model's on-policy data can be as effective as target model data. GKD achieves the best wall-time performance improvement on XSum. (b) Distillation improves the block efficiency for all examples on GSM8K. (c) Empirical block efficiency aligns well with its $D_{\text{TVD}}$-based theoretical counterpart.

generated by the target model, suggests that white-box distillation (i.e., supervision from the target model's logits) is vital for SD. This is corroborated by Figure 3a, which illustrates the evolution of the acceptance rate throughout training. Supervised KD ranks lowest, and its performance plateaus early during training due to the static nature of the dataset. In contrast, algorithms using model-generated data lead to continual improvement of the acceptance rate. Despite $f$-Distill being much more computationally costly than GKD due to the use of teacher-generated data, both algorithms exhibit comparable performance. Notably, GKD achieves the best wall-time performance improvement. See Appendix C.1.2 for more visualizations of performance improvement during training.

We also investigate whether KD improves block efficiency universally or impacts a limited subset of examples. Figure 3b depicts the improvement per example. We observe consistent gains in block efficiency across most examples, which can also be seen in various datasets (see Figure C.16). Figure 3c illustrates a strong agreement between theoretical and empirical block efficiency values for several distilled models (each model is represented as a filled circle). Despite theoretical values occasionally overestimating or underestimating empirical values, possibly due to potential deviations from the i.i.d. token-level assumption (cf. §3), the ranking of distilled models remains highly consistent. In summary, these findings largely confirm that KD effectively optimizes block efficiency.

**Transferability of distilled models.** We next examine the transferability of distilled models on diverse datasets unseen during training. We use a draft model distilled on GSM8K and test its ability to speed up SD on zero-shot chain-of-thought (CoT) prompting over 23 reasoning tasks from the BigBenchHard suite (Suzgun et al., 2022). The results, illustrated in Figure 1, indicate effective transferability to other datasets. Compared to standard SD, the distilled model significantly enhances average decoding speeds, yielding speedup improvements from $1.93\times$ and $1.78\times$ to $2.21\times$ and $2.02\times$ for greedy and non-greedy decoding methods, respectively. Further analysis in Figure C.1 reveals that using our distilled T5-Small as draft model is also compatible with larger target models (T5-XXL) despite being distilled from a different-sized model (T5-XL). Despite being not fully optimized, this configuration consistently outperforms standard SD by $7\% - 37\%$ across various datasets. See Appendix C.1 for more details.

## 5.2 DISTILLSPEC RECIPE

We now focus on identifying the optimal KD approach for SD. Following the training and evaluation protocols in § 5.1, we explore four training data construction methods and four divergence functions on XSum and GSM8K. Specifically, we explore the following variants of training data: 1) fixed ground-truth dataset $\mathcal{D}_{\text{Train}}$, 2) data generated only from the draft model $\mathcal{M}_q^\theta$, 3) data generated only from target $\mathcal{M}_p$, 4) data generated from both $\mathcal{M}_q^\theta$ and $\mathcal{M}_p$ in equal proportion. We also examine the following divergences: 1) forward KL (FKL), 2) Jenson-Shannon divergence (JSD), 3) reverse KL (RKL), and 4) total variation distance (TVD).

**Importance of training data and divergence in DistillSpec.** Figure 4 illustrates the block efficiency improvement on XSum and GSM8K, in line with observations from § 5.1. We note that using model-generated data (last three rows) yields superior performance than using a fixed dataset (first row). Specifically, on XSum with greedy decoding, using data generated from both $\mathcal{M}_q^\theta$ and $\mathcal{M}_p$ leads to the best performance, with JSD slightly outperforming the other divergences. However,

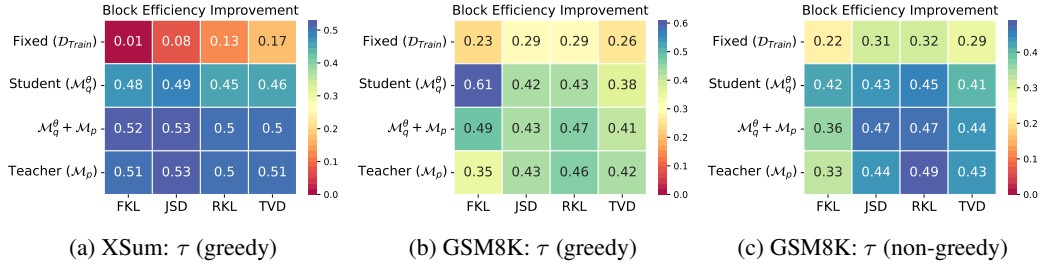

Figure 4: **DistillSpec recipe**. We report improvement in empirical block efficiency post-distillation on (a) XSum with greedy sampling, and GSM8K with (b) greedy and (c) non-greedy sampling. Which divergence function and data construction lead to the largest block efficiency improvements highly depends on the task. They should be treated as a hyperparameters to be tuned on our task of interest.

on GSM8K with greedy decoding, FKL with only draft-generated data emerges as the best setup. In contrast, with temperature sampling (at $T = 1$), a different trend is observed as RKL combined with data generated by $\mathcal{M}_p$ is the most effective setup. See Appendix C.2.1 for results on different datasets and decoding strategies. Nonetheless, using only draft-generated data is found to be competitive.

**Impact of distillation on draft quality vs. compatibility.** We also study how different distillation recipes affect task performance and whether there is any one design choice that is simultaneously optimal for improving both draft model task performance and its utility for SD (cf. Figure C.23, C.24). Similar to our earlier observations, the use of generated data is paramount for improving draft performance. Notably, utilizing data generated from $\mathcal{M}_q^\theta$ yields comparable or superior results compared to using data generated from $\mathcal{M}_p$. However, which KD algorithm is optimal largely depends on the task at hand and the underlying decoding strategy. Figure 5a highlights an interesting dichotomy between block efficiency improvements and task performance gains via KD: distilled models with high task performance do not necessarily translate into more aligned drafters for SD.

**Recommendation.** Interestingly, although TVD is the objective we should aim to optimize for SD (cf. Eq. 3), its direct optimization does not yield the best performance in most of the settings explored. We generally find that the choice of divergence in KD is a hyperparameter that needs to be tuned based on the task at hand and decoding strategy used. For training data construction, we propose using the draft model $\mathcal{M}_q^\theta$ for data generation as it can achieve similar or superior performance compared to the target model $\mathcal{M}_p$, but at a much lower cost.

## 5.3 QUALITY VERSUS LATENCY TRADE-OFF

**Lossy speculative decoding.** We analyze the quality-latency trade-off using lossy SD variants, as detailed in Algorithm A.1. As Figure 5b illustrates, employing either KD ($\star$) or SD ($\times$) alone does not fully bridge the performance or latency gaps, respectively. In such case, a leniency parameter ($\varepsilon$) can help interpolate between these two approaches, as demonstrated in Figure 5b where each point within a given group corresponds to a different value of $\varepsilon$. As the GSM8K experiment shows, the power of interpolation can be limited: even using a permissive lenience of $\varepsilon = 10^{-5}$, $f_{\mathrm{lin}}$ still results in high performance but high latency, while $f_{\mathrm{sq}}$ traces a similar but slightly extended trade-off curve. Although $f_{\mathrm{exp}}$ makes interpolation possible, it yields a worse quality-latency trade-off. Interestingly, it is possible to significantly reduce latency while preserving most of the quality on GSM8K, possibly because many tokens are inconsequential for final performance, and a variety of proposals can be safely accepted with minimal effect on generation quality. See Appendix C.3.1 for a comparison between non-distilled and distilled draft models, where we show that a distilled draft model enables a much better quality vs. latency trade-off.

**DistillSpec meets model garden.** In many practical scenarios, we have access to multiple models of different sizes—a model garden—to design the inference pipeline. We emulate this setting by focusing on the five model sizes in the T5 model family: T5-Small (77M), T5-Base (250M), T5-Large (800M), T5-XL (3B), and T5-XXL (11B). We study the quality-latency trade-off curves obtained from applying KD and SD as follows: 1) **raw:** deploying supervised fine-tuned (SFT) T5 models; 2) **distilled:** applying KD by distilling smaller models from the larger T5 models; 3) **speculative:** applying SD using T5 models; and 4) **DistillSpec:** applying KD on T5 models and using SD with distilled models as target and draft models.

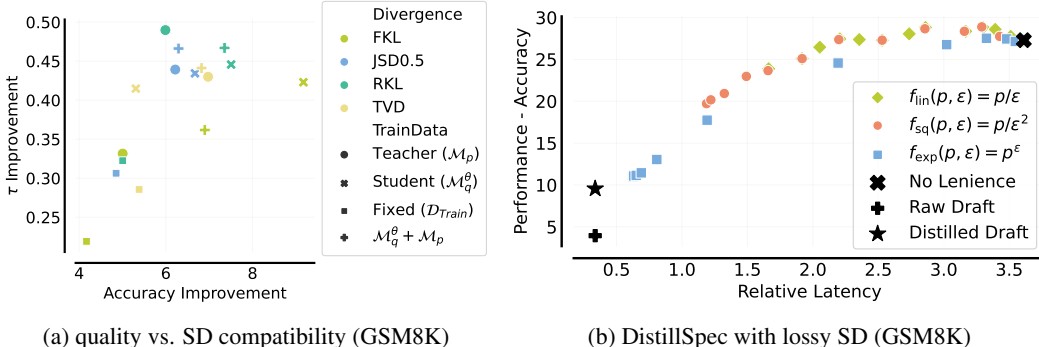

(a) quality vs. SD compatibility (GSM8K)      (b) DistillSpec with lossy SD (GSM8K)

Figure 5: (a) The improvement on speculative decoding and downstream task performance are only weakly correlated. A high quality distilled model does not imply it can be an effective draft model in speculative decoding. (b) We employ leniency as a precise control mechanism to achieve the desired quality-latency profile.

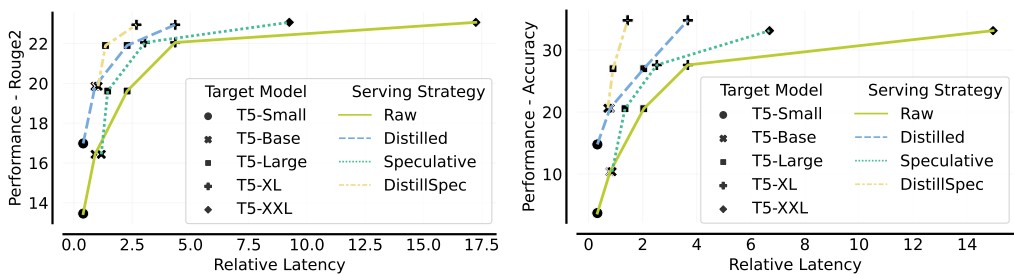

Figure 6: DistillSpec excels in both quality and latency, offering a remarkable 6.4x and 10.7x latency reduction on XSum (left) and GSM8K (right), while maintaining nearly identical performance.

Figure 6 shows that SD effectively shifts the trade-off curve leftward, especially with larger model sizes. However, its efficacy diminishes with smaller model sizes when the computation time between the draft and target models is closely matched. In contrast, distillation, which optimizes the model for downstream task performance, appears to offer a superior trade-off between quality and latency, particularly for smaller models. Conversely, a reverse trend is observed for larger model sizes when evaluating the model with temperature sampling. Figure C.30a indicates a substantial gap between the distilled model and the larger teacher model, while the SD-based method is able to significantly reduce latency. This suggests that when stringent performance and decoding strategy constraints are in place, SD remains a valuable approach. Our DistillSpec method, which combines the benefits of distillation and SD, consistently achieves the best trade-off between quality and latency, yielding an impressive reduction in latency while maintaining nearly identical performance. Specifically, DistillSpec reduces relative latency from 17.3 to 2.7 and from 15.0 to 1.4 on XSum and GSM8K, respectively, representing speedup improvements of $6.4\times$ and $10.7\times$. In contrast, the Rouge2 score only marginally decreases, from 23.1 to 23.0 on XSum, while the model accuracy on GSM8K actually improves from 33.1 to 34.8.

# 6 CONCLUSION

In this paper, we evaluate the efficacy of white-box knowledge distillation (KD) in enhancing speculative decoding (SD) through improved alignment between target and draft models. A thorough analysis is conducted to understand the impact of training data construction and divergence functions on KD performance. We underscore the significance of utilizing model-generated data and argue that employing the draft model's on-policy data during KD is a cost-efficient way of improving model alignment. Additionally, we assess the trade-off between quality and latency within the scope of lenience and availability of multiple models of varying quality and size (model garden), concluding that KD enables a superior trade-off compared to standard SD. The optimal strategy involves first applying KD for downstream task performance, followed by SD using a distilled draft model, resulting in a six to ten-fold decrease in latency with negligible performance loss. Our study contributes novel insights into white-box KD algorithms for LLMs and provides guidance for striking an effective balance between quality and latency using KD and SD.

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

# Appendix

## Table of Contents

# A  METHOD

## A.1  DESCRIPTION OF DIVERGENCE FUNCTIONS

Below are some common divergence functions used in distillation, given two discrete probability distribution $P(\mathcal{C})$ and $Q(\mathcal{C})$.

**Kullback-Leibler (KL) divergence.**

$$D_{\mathrm{KL}}(P\|Q) = \sum_{c \in \mathcal{C}} P(c) \log \frac{P(c)}{Q(c)} \tag{A.1}$$

We note that the KL divergence is not symmetric, i.e., $D_{\mathrm{KL}}(P\|Q) \neq D_{\mathrm{KL}}(Q\|P)$. As such, we refer to $D_{\mathrm{KL}}(P\|Q)$ as the **forward KL** (FKL) and $D_{\mathrm{RKL}}(P\|Q) := D_{\mathrm{KL}}(Q\|P)$ as the **reverse KL** (RKL) between $P$ and $Q$. Minimization of the FKL under an empirical data distribution corresponds to maximum likelihood estimation (**MLE**), which is the typical loss objective used in supervised learning given a fixed dataset.

**Jensen–Shannon (JS) divergence.**

$$D_{\mathrm{JS}}(P\|Q) = \frac{1}{2}(D_{\mathrm{KL}}(P\|M) + D_{\mathrm{KL}}(Q\|M)), \quad \text{where } M = \frac{1}{2}(P + Q) \tag{A.2}$$

**Generalized Jensen-Shannon divergence ($D_{\mathrm{JSD}[\beta]}$).**

$$D_{\mathrm{JSD}[\beta]}(P\|Q) = \beta D_{\mathrm{KL}}\Big(P\Big\|\beta P + (1 - \beta)Q\Big) + (1 - \beta)D_{\mathrm{KL}}\Big(Q\Big\|\beta P + (1 - \beta)Q\Big) \tag{A.3}$$

Interestingly, it can be proved that $\lim_{\beta \to 0} \frac{D_{JSD(\beta)}(P\|Q)}{\beta} = D_{\mathrm{KL}}(P\|Q)$ (Huszár, 2015). As such, $D_{\mathrm{JSD}[\beta]}$ behaves similarly to forward KL for small values of $\beta$. Similarly, $D_{\mathrm{JSD}[\beta]}$ has similar behavior to reverse KL for $\beta$ close to 1, since $D_{\mathrm{JSD}[\beta]}(P\|Q) = D_{\mathrm{JSD}[1-\beta]}(Q\|P)$.

**Total variation distance (TVD).**

$$D_{\mathrm{TVD}}(P\|Q) = \sum_{c \in \mathcal{C}} |\frac{P(c) - Q(c)}{2}| \tag{A.4}$$

## A.2 JUSTIFICATION OF USING ON-POLICY DATA

In this section, we prove Theorem 4.1 which motivates our use of on-policy data. We follow the notations in Sections 3 and 4. In addition, we write $\epsilon(x) := \mathbb{E}_{y \sim q_{\leq T}(y \mid x)} \left[ D_{\text{TVD}}(\mathcal{M}_p \| \mathcal{M}_q^\theta)(y \mid x) \right]$ for the distillation loss of a single input $x$.

To ease the analysis, let decompose $\alpha(x)$ and $\epsilon(x)$ into sums of contributions from each token.

**Lemma A.1.** *For all $x$, $\alpha(x) = 1 - \frac{1}{L_p(x)} \sum_{t=1}^T A_t$ and $\epsilon(x) \geq \frac{1}{T} \sum_{t=1}^T E_t$, where:*

$$A_t(x) := \mathbb{E}_{y \sim p_{\leq T}(y \mid x)} \left[ \mathbb{1}_{\{t \leq |y|\}} D_{\text{TVD}}(p(y_t \mid x, y_{<t}), q(y_t \mid x, y_{<t})) \right], \tag{A.5}$$

$$E_t(x) := \mathbb{E}_{y \sim q_{\leq T}(y \mid x)} \left[ \mathbb{1}_{\{t \leq |y|\}} D_{\text{TVD}}(p(y_t \mid x, y_{<t}), q(y_t \mid x, y_{<t})) \right]. \tag{A.6}$$

*Proof.* By Equation 3, we have:

$$\alpha(x) = 1 - \frac{\mathbb{E}_{y \sim p_{\leq T}(y \mid x)} \left[ \sum_{t=1}^{|y|} D_{\text{TVD}}(p(y_t \mid x, y_{<t}), q(y_t \mid x, y_{<t})) \right]}{L_p(x)}.$$

We can rewrite $\sum_{t=1}^{|y|} D_{\text{TVD}}(p(y_t), q(y_t)) = \sum_{t=1}^T \mathbb{1}_{\{t \leq |y|\}} D_{\text{TVD}}(p(y_t), q(y_t))$ and swap the order between the summation and the expectation to obtain:

$$\alpha(x) = 1 - \frac{\sum_{t=1}^T \mathbb{E}_{y \sim p_{\leq T}(y \mid x)} \left[ \mathbb{1}_{\{t \leq |y|\}} D_{\text{TVD}}(p(y_t \mid x, y_{<t}), q(y_t \mid x, y_{<t})) \right]}{L_p(x)},$$

which proves Equation A.5.

Similarly, by definition of $\epsilon(x)$ and $D_{\text{TVD}}$ we have:

$$\epsilon(x) = \mathbb{E}_{y \sim q_{\leq T}(y \mid x)} \left[ \frac{1}{|y|} \sum_{t=1}^{|y|} D_{\text{TVD}}(p(y_t \mid x, y_{<t}), q(y_t \mid x, y_{<t})) \right]$$

$$\geq \mathbb{E}_{y \sim q_{\leq T}(y \mid x)} \left[ \frac{1}{T} \sum_{t=1}^{|y|} D_{\text{TVD}}(p(y_t \mid x, y_{<t}), q(y_t \mid x, y_{<t})) \right].$$

Again, we can rewrite $\sum_{t=1}^{|y|} D_{\text{TVD}}(p(y_t), q(y_t)) = \sum_{t=1}^T \mathbb{1}_{\{t \leq |y|\}} D_{\text{TVD}}(p(y_t), q(y_t))$ and then swap the order between the summation and the expectation to obtain:

$$\epsilon(x) \geq \frac{1}{T} \sum_{t=1}^T \mathbb{E}_{y \sim q_{\leq T}(y \mid x)} \left[ \mathbb{1}_{\{t \leq |y|\}} D_{\text{TVD}}(p(y_t \mid x, y_{<t}), q(y_t \mid x, y_{<t})) \right],$$

which proves Equation A.6. □

Lemma A.1 motivates us to study $A_t(x)$ and $E_t(x)$ instead. Below we rewrite them in variational forms that will be used later. For this, we introduce some defintiions.

**Definition A.1.** *For any sequence $z \in \{\text{P}, \text{Q}\}^\tau$ that consists only of letters P and Q, we define $\mathcal{M}(x, y, z)$ as the distribution of sequences sampled as follows:*

1. *Initialize a sequence of tokens as $y$;*

2. *If there are $t - 1$ tokens, sample the $t$-th token from $\mathcal{M}_p$ if $z_t = \text{P}$, and from $\mathcal{M}_q$ otherwise;*

3. *Repeat until an end-of-sequence token is sampled, or the sequence length has reached $\tau$.*

We use the shorthand $\text{P}^k$ and $\text{Q}^k$ to represent the sequence of $k$ consecutive letters of P and Q respectively. Let $\Omega^k$ denote the set of all possible strings of length $k$, and $\delta : \Omega^t \to [-1/2, 1/2]$ be a generic function that maps a sequence of $t$ tokens to a real number in $[-1, 1]$. We abuse the notation and assign $\delta(y) = 0$ for all $y \notin \Omega^t$.

**Lemma A.2.** *For all $x$ and all $1 \leq t \leq T$:*

$$A_t(x) = \sup_{\delta: \Omega^t \to [-1/2, 1/2]} \left\{ \mathbb{E}_{y \sim \mathcal{M}(x, \varnothing, \mathtt{P}^t)} \left[ \delta(y) \right] - \mathbb{E}_{y \sim \mathcal{M}(x, \varnothing, \mathtt{P}^{t-1}\mathtt{Q})} \left[ \delta(y) \right] \right\}, \tag{A.7}$$

$$E_t(x) = \sup_{\delta: \Omega^t \to [-1/2, 1/2]} \left\{ \mathbb{E}_{y \sim \mathcal{M}(x, \varnothing, \mathtt{Q}^{t-1}\mathtt{P})} \left[ \delta(y) \right] - \mathbb{E}_{y \sim \mathcal{M}(x, \varnothing, \mathtt{Q}^t)} \left[ \delta(y) \right] \right\}. \tag{A.8}$$

*Proof.* For a fixed pair $x$ and $y$, we rewrite the total variance distance between $p(y_t \,|\, x, y_{<t})$ and $q(y_t \,|\, x, y_{<t})$ as the following variational form:

$$D_{\text{TVD}}(p(y_t \,|\, x, y_{<t}), q(y_t \,|\, x, y_{<t}))$$

$$= \sup_{\tilde{\delta}: \Omega \to [-1/2, 1/2]} \left\{ \underbrace{\mathbb{E}_{y_t \sim p(y_t \,|\, x, y_{<t})} \left[ \tilde{\delta}(y_t) \right] - \mathbb{E}_{y_t \sim q(y_t \,|\, x, y_{<t})} \left[ \tilde{\delta}(y_t) \right]}_{=: \Delta(x, y_{<t}, \tilde{\delta})} \right\}.$$

After taking the expectations:

$$A_t(x) = \mathbb{E}_{y \sim p_{\leq T}(y \,|\, x)} \left[ D_{\text{TVD}}(p(y_t \,|\, x, y_{<t}), q(y_t \,|\, x, y_{<t})) \right]$$

$$= \mathbb{E}_{y \sim p_{\leq T}(y \,|\, x)} \left[ \sup_{\tilde{\delta}: \Omega \to [-1/2, 1/2]} \left\{ \Delta(x, y_{<t}, \tilde{\delta}) \right\} \right]$$

$$= \mathbb{E}_{y \sim \mathcal{M}(x, \varnothing, \mathtt{P}^{t-1})} \left[ \sup_{\tilde{\delta}: \Omega \to [-1/2, 1/2]} \left\{ \mathbb{1}_{\{|y|=t-1\}} \Delta(x, y, \tilde{\delta}) \right\} \right]$$

$$= \sup_{\delta: \Omega^t \to [-1/2, 1/2]} \left\{ \mathbb{E}_{y \sim \mathcal{M}(x, \varnothing, \mathtt{P}^{t-1})} \left[ \mathbb{1}_{\{|y|=t-1\}} \Delta(x, y, \delta(y, \cdot)) \right] \right\},$$

where the third equality uses the observation that for any function $f$, $\mathbb{E}_{y \sim p_{\leq T}(y \,|\, x)} \left[ f(x, y_{<t}) \right]$ can be replaced with $\mathbb{E}_{y \sim \mathcal{M}(x, \varnothing, \mathtt{P}^{t-1})} \left[ \mathbb{1}_{\{|y|=t-1\}} f(x, y) \right]$, and the last equality swaps the order between the expectation and the supremum. Finally, we move the expectations in $\Delta$ to merge with the expectation outside and obtain:

$$A_t(x) = \sup_{\delta: \Omega^t \to [-1/2, 1/2]} \left\{ \mathbb{E}_{y \sim \mathcal{M}(x, \varnothing, \mathtt{P}^t)} \left[ \mathbb{1}_{\{|y|=t\}} \delta(y) \right] - \mathbb{E}_{y \sim \mathcal{M}(x, \varnothing, \mathtt{P}^{t-1}\mathtt{Q})} \left[ \mathbb{1}_{\{|y|=t\}} \delta(y) \right] \right\}$$

$$= \sup_{\delta: \Omega^t \to [-1/2, 1/2]} \left\{ \mathbb{E}_{y \sim \mathcal{M}(x, \varnothing, \mathtt{P}^t)} \left[ \delta(y) \right] - \mathbb{E}_{y \sim \mathcal{M}(x, \varnothing, \mathtt{P}^{t-1}\mathtt{Q})} \left[ \delta(y) \right] \right\},$$

where the last step follows from our abuse of notation that $\delta(y) = 0$ for all $y \notin \Omega^t$. This proves Equation A.7, and Equation A.8 can be proved similarly. $\qquad\square$

Based on our variational forms of $A_t(x)$ and $E_t(x)$, we obtain the following lemma for bounding $A_t(x)$ in terms of $E_t(x)$.

**Lemma A.3.** *For all $x$ and $1 \leq t \leq T$:*

$$A_t(x) \leq 2 \sum_{k=1}^{t-1} E_k(x) + E_t(x). \tag{A.9}$$

*Proof.* It suffices to find an upper bound on $\mathbb{E}_{y \sim \mathcal{M}(x, \varnothing, \mathtt{P}^t)} \left[ \delta(y) \right]$ and a lower bound on $\mathbb{E}_{y \sim \mathcal{M}(x, \varnothing, \mathtt{P}^{t-1}\mathtt{Q})} \left[ \delta(y) \right]$ for all $\delta: \Omega^t \to [-1/2, 1/2]$.

For all $1 \leq k \leq t$, we can replace the first $\mathtt{P}$ in $\mathbb{E}_{y \sim \mathcal{M}(x, \varnothing, \mathtt{Q}^{k-1}\mathtt{P}^{t-k+1})} \left[ \delta(y) \right]$ with $\mathtt{Q}$ by only introducing an error of $E_{k+1}(x)$:

$$\mathbb{E}_{y \sim \mathcal{M}(x, \varnothing, \mathtt{Q}^{k-1}\mathtt{P}^{t-k+1})} \left[ \delta(y) \right] = \mathbb{E}_{y \sim \mathcal{M}(x, \varnothing, \mathtt{Q}^{k-1}\mathtt{P})} \left[ \mathbb{1}_{\{|y|=k\}} \mathbb{E}_{y' \sim \mathcal{M}(x, y, \mathtt{P}^{t-k})} \left[ \delta(y') \right] \right]$$

$$\leq \mathbb{E}_{y \sim \mathcal{M}(x, \varnothing, \mathtt{Q}^k)} \left[ \mathbb{1}_{\{|y|=k\}} \mathbb{E}_{y' \sim \mathcal{M}(x, y, \mathtt{P}^{t-k})} \left[ \delta(y') \right] \right] + E_k(x)$$

$$= \mathbb{E}_{y \sim \mathcal{M}(x, \varnothing, \mathtt{Q}^k \mathtt{P}^{t-k})} \left[ \delta(y) \right] + E_k(x),$$

where the second inequality holds because we can define a function $\delta' : \Omega^k \to [-1/2, 1/2]$, $such\,that\,\delta(y) = \mathbb{1}_{\{|y|=k\}}\mathbb{E}_{y' \sim \mathcal{M}(x,y,\mathtt{P}^{t-k})}[\delta(y')]$ and then apply Equation A.8.

Now taking a telescoping sum over $1 \leq k \leq t$, we obtain:

$$\mathbb{E}_{y \sim \mathcal{M}(x,\varnothing,\mathtt{P}^t)}[\delta(y)] \leq \mathbb{E}_{y \sim \mathcal{M}(x,\varnothing,\mathtt{Q}^t)}[\delta(y)] + \sum_{k=1}^{t} E_k(x). \tag{A.10}$$

Similarly, for all $1 \leq k \leq t-1$, we can replace the first $\mathtt{P}$ in $\mathbb{E}_{y \sim \mathcal{M}(x,\varnothing,\mathtt{Q}^{k-1}\mathtt{P}^{t-k}\mathtt{Q})}[\delta(y)]$ with $\mathtt{Q}$ and only introduce an error of $E_k(x)$:

$$\begin{aligned}
\mathbb{E}_{y \sim \mathcal{M}(x,\varnothing,\mathtt{Q}^{k-1}\mathtt{P}^{t-k}\mathtt{Q})}[\delta(y)] &= \mathbb{E}_{y \sim \mathcal{M}(x,\varnothing,\mathtt{Q}^{k-1}\mathtt{P})}\left[\mathbb{1}_{\{|y|=k\}}\mathbb{E}_{y' \sim \mathcal{M}(x,y,\mathtt{P}^{t-k-1}\mathtt{Q})}[\delta(y')]\right] \\
&= -\mathbb{E}_{y \sim \mathcal{M}(x,\varnothing,\mathtt{Q}^{k-1}\mathtt{P})}\left[-\mathbb{1}_{\{|y|=k\}}\mathbb{E}_{y' \sim \mathcal{M}(x,y,\mathtt{P}^{t-k-1}\mathtt{Q})}[\delta(y')]\right] \\
&\geq -\left(\mathbb{E}_{y \sim \mathcal{M}(x,\varnothing,\mathtt{Q}^k)}\left[-\mathbb{1}_{\{|y|=k\}}\mathbb{E}_{y' \sim \mathcal{M}(x,y,\mathtt{P}^{t-k-1}\mathtt{Q})}[\delta(y')]\right] + E_k(x)\right) \\
&= \mathbb{E}_{y \sim \mathcal{M}(x,\varnothing,\mathtt{Q}^k\mathtt{P}^{t-k-1}\mathtt{Q})}[\delta(y)] - E_k(x).
\end{aligned}$$

Taking a telescoping sum over $1 \leq k \leq t-1$ yields:

$$\mathbb{E}_{y \sim \mathcal{M}(x,\varnothing,\mathtt{P}^{t-1}\mathtt{Q})}[\delta(y)] \geq \mathbb{E}_{y \sim \mathcal{M}(x,\varnothing,\mathtt{Q}^t)}[\delta(y)] - \sum_{k=1}^{t-1} E_k(x). \tag{A.11}$$

Subtracting Equation (A.11) from Equation (A.10), we have the following holds for all functions $\delta : \mathbb{S} \to [-1/2, 1/2]$:

$$\mathbb{E}_{y \sim \mathcal{M}(x,\varnothing,\mathtt{P}^t)}[\delta(y)] - \mathbb{E}_{y \sim \mathcal{M}(x,\varnothing,\mathtt{P}^{t-1}\mathtt{Q})}[\delta(y)] \leq \sum_{k=1}^{t} E_k(x) + \sum_{k=1}^{t-1} E_k(x) = 2\sum_{k=1}^{t-1} E_k(x) + E_t(x),$$

which proves the claim. $\qquad \square$

*Proof of Theorem 4.1.* Summing Equation A.9 over $1 \leq t \leq T$, we have:

$$\sum_{t=1}^{T} A_t(x) \leq \sum_{t=1}^{T}(1 + 2(T-t))E_t(x).$$

Combining this with Lemma A.1 yields:

$$\begin{aligned}
\alpha(x) = 1 - \frac{1}{L_p(x)}\sum_{t=1}^{T} A_t &\geq 1 - \frac{1}{L_p(x)}\sum_{t=1}^{T}(1 + 2(T-t))E_t(x) \\
&\geq 1 - \frac{2T}{L_p(x)}\sum_{t=1}^{T} E_t(x) \\
&\geq 1 - \frac{2T^2}{L_p(x)}\epsilon(x),
\end{aligned}$$

which proves the theorem statement after taking the expectation over $x \sim X$. $\qquad \square$

## A.3 DISTILLSPEC ALGORITHMS

---

**Algorithm A.1** Speculative decoding step

---

**Require:** target model $\mathcal{M}_p$, draft model $\mathcal{M}_q$, context $\rho = \{x, y_{<t}\}$.
**Hyperparameters:** block size ($\gamma$), lenience Function $f(p, \epsilon)$ ($\epsilon = 1$ for lossless and $\epsilon < 1$ for lossy decoding).

1: **for all** $i = 0$ to $\gamma - 1$ **do**
2:      $q_{t+i}(y) \leftarrow \mathcal{M}_q(x, y_{<t+i}), \quad y_{t+i} \sim q_{t+i}(y)$          $\triangleright$ Sample $\gamma$ tokens from $\mathcal{M}_q$ autoregressively.
3: **end for**
4: $(p_t(y), \ldots, p_{t+\gamma}(y)) \leftarrow (\mathcal{M}_p(x, y_{<t}), \ldots, \mathcal{M}_p(x, y_{<t+\gamma}))$          $\triangleright$ Run $\mathcal{M}_p$ in parallel.
5: $r_i \leftarrow \frac{f(p_i(y), \epsilon)}{q_i(y)}, \quad \forall t \le i < t + \gamma$          $\triangleright$ Compute the rejection thresholds.
6: $u_t \sim \text{Uniform}(0, 1), \ldots, u_{t+\gamma-1} \sim \text{Uniform}(0, 1)$          $\triangleright$ Generate $\gamma$ random values.
7: $n \leftarrow \min(\{i \mid 0 \le i < \gamma, u_{t+i} > r_{t+i}\} \cup \{\gamma\})$          $\triangleright$ Determine the number of accepted tokens $n$.
8: **if** $n < \gamma$ **then**
9:      $y_{t+n} \sim \text{norm}(\max(0, p_{t+n}(y) - q_{t+n}(y)))$          $\triangleright$ Sample from the adjusted distribution.
10: **else**
11:      $y_{t+n} \sim p_{t+n}(y)$          $\triangleright$ Sample from $\mathcal{M}_p$.
12: **end if**
     **Return** $\{x, y_{<t+n+1}\}$          $\triangleright$ Append $n$ tokens from $\mathcal{M}_q$ and one token from $\mathcal{M}_p$.

---

**Algorithm A.2** Knowledge distillation

---

**Require:** target model $\mathcal{M}_p$, draft model $\mathcal{M}_q^\theta$, dataset $(X, Y)$ containing input $x$ and possibly output $y$.
**Hyperparameters:** fixed data fraction $\lambda_1 \in [0, 1]$, student data fraction $\lambda_2 \in [0, 1]$, divergence function $D$, learning rate $\eta$.

1: $u_1 \sim \text{Uniform}(0, 1), u_2 \sim \text{Uniform}(0, 1)$          $\triangleright$ Generate two random values.
2: **if** $u_1 \le \lambda_1$ **then**
3:      $B = \{(x_b, y_b)\}_{b=1}^B$, where $(x_i, y_i) \sim (X, Y)$          $\triangleright$ Sample inputs and outputs from $(X, Y)$.
4: **else**
5:      $B' = \{(x_b)\}_{b=1}^B$, where $x_i \sim X$          $\triangleright$ Sample a batch of inputs from $X$.
6:      **if** $u_2 \le \lambda_2$ **then**
7:          $B = \{(x_b, y_b)\}_{b=1}^B$, where $x_i \sim B', y_i \sim \mathcal{M}_q^\theta(\cdot | x_i)$          $\triangleright$ Sample data from $\mathcal{M}_q$.
8:      **else**
9:          $B = \{(x_b, y_b)\}_{b=1}^B$, where $x_i \sim B', y_i \sim \mathcal{M}_p(\cdot | x_i)$          $\triangleright$ Sample data from $\mathcal{M}_p$.
10:      **end if**
11: **end if**
     **Return** $\theta \leftarrow \theta - \eta \frac{1}{B} \sum_{(x,y) \in B} \nabla_\theta D(\mathcal{M}_p \| \mathcal{M}_q^\theta)(y | x)$          $\triangleright$ Update $\theta$ to minimize $D(\mathcal{M}_p \| \mathcal{M}_q^\theta)$.

---

# B  IMPLEMENTATION DETAILS

## B.1  DATASETS

In this section, we present a comprehensive overview of the datasets employed in this study.

**XSum (Narayan et al., 2018).**    The Extreme Summarization (XSum) dataset serves as an evaluation benchmark for abstractive single-document summarization systems. This dataset comprises 226,711 news articles, sourced from BBC articles spanning the years 2010 to 2017. These articles encompass a wide range of domains, including News, Politics, Sports, Weather, Business, Technology, Science, Health, Family, Education, Entertainment, and Arts. Summarization performance is evaluated using ROUGE scores on the validation dataset split of XSum and primarily emphasizes ROUGE-2, but with similar trends observed for ROUGE-LSum and ROUGE-1. A maximum input length of 1,024 and a maximum output length of 64 are employed for distillation and evaluation.

**CNN/DM (Hermann et al., 2015).**    The CNN/Daily Mail (CNN/DM) dataset is tailored for text summarization. It comprises abstractive summary bullets generated by humans from news stories on CNN and Daily Mail websites, presented in the form of questions with entities hidden. The questions can be answered using relevant passages from the source text. Similar to XSum, ROUGE scores on the validation dataset are reported, primarily emphasizing ROUGE-2 but with similar trends observed for ROUGE-LSum and ROUGE-1. A maximum input length of 2,048 and a maximum output length of 128 are used for distillation and evaluation.

**WMT EnDe (Bojar et al., 2014).**    The WMT14 EnDe dataset stands as a standard benchmark for machine translation. The task entails translating English text to German while preserving content, semantics and style. Evaluation is based on BLEU scores, which measures the similarity of machine-translated text to high-quality reference translations. A maximum input length of 80 and a maximum output length of 80 are employed for distillation and evaluation, with performance assessed on the original test split.

**GSM8K (Cobbe et al., 2021).**    The GSM8K dataset comprises 8.5K high-quality, linguistically diverse grade school math word problems crafted by human problem writers. The dataset is divided into 7.5K training problems and 1K test problems, with solutions typically requiring 2 to 8 steps involving elementary calculations using basic arithmetic operations. To enhance reasoning abilities, we explored distillation alongside zero-shot chain-of-thought (CoT), as described in Agarwal et al. (2023). A maximum input length of 256 and a maximum output length of 320 are used for distillation and evaluation.

**LM1B Chelba et al. (2013).**    The One Billion Word dataset (LM1B) is a widely recognized benchmark for language modeling. The training and held-out data are derived from the WMT 2011 News Crawl dataset using Bash shell and Perl scripts. A maximum input length of 128 and a maximum output length of 128 are used for distillation and evaluation.

## B.2  MODELS

In accordance with Leviathan et al. (2023), we evaluate two model types: 1) GPT-like decoder-only Transformer models trained on the LM1B task (Chelba et al., 2013) using the standard autoregressive objective, where the target and draft models have 234M and 33M parameters, respectively; and 2) standard encoder-decoder T5 v1.1 models (Raffel et al., 2020) supervised fine-tuned on four different tasks, with T5-XL (3B) and T5-Small (77M) serving as the target and draft models, respectively.

The target $\mathcal{M}_p$ in the decoder-only model experiment has hidden dimension 1,024, feed-forward dimension 4,096, 12 layers and 16 attention heads per transformer block, for a total of 234M parameters. The corresponding draft model $\mathcal{M}_q^\theta$ has hidden dimension 512, feed-forward dimension 1,024, 4 layers and 4 attention heads per transformer block, for a total of 33M parameters. All models utilize the T5 tokenizer with 32k tokens. As for the T5 base checkpoints, we start from LM-adapted T5v1.1 models. These LM-adapted models are initialized from T5v1.1 and trained for an additional 100K steps on the LM objective discussed in the T5 paper (Raffel et al., 2020). These checkpoints are available at `https://console.cloud.google.com/storage/browser/t5-data/pretrained_models`.

In our encoder-decoder distillation experiments, both the student and teacher models are initialized from models supervised fine-tuned on the original training dataset. This process for each dataset is detailed as follows:

- **XSum**: for T5-Small, -Base, -Large, -XL and -XXL models, we warm start distillation from LM-Adapted T5v1.1 models supervised fine-tuned for 100K, 50K, 30k, 20K and 8K steps, respectively.

- **CNN/DM**: for T5-Small,-Base,-Large, -XL and -XXL models, we warm start distillation from LM-Adapted T5v1.1 models supervised fine-tuned for 200K, 80K, 20k, 20k and 4K steps, respectively.

- **WMT**: for T5-Small,-Base,-Large, -XL and -XXL models, we warm start distillation from LM-Adapted T5v1.1 models supervised fine-tuned for 250K, 250K, 110k , 50K and 10K steps, respectively.

- **GSM8K**: all models are supervised fine-tuned starting from FLAN-T5 models on the PaLM-540 generated CoT dataset for 10K steps.

### B.3 DISTILLATION

**Training Data for KD.** We study the five KD algorithms outlined in Table 1. For SeqKD (Kim & Rush, 2016) and $f$-Distill (Wen et al., 2023), we opt for an online teacher data generation approach instead of a conventional fixed offline teacher-generated dataset. This approach, while computationally expensive, yields a more diverse dataset. For GKD, we exclude the static ground truth data and solely rely on the data generated by the online student model. All data generated by the teacher or the student is based on temperature sampling with a temperature of 1.0 (see Appendix C.1.3 for an ablation study on sampling temperature).

**Training Details.** We employ an Adafactor optimizer (Shazeer & Stern, 2018) to train the draft student model ($\mathcal{M}_q^\theta$) in all our experiments, following the guidelines outlined in Algorithm A.2. In the context of our knowledge distillation (KD) loss function, defined in Eq. 2, we maintain the temperatures for both the target model, denoted as $T_p$, and the draft model, denoted as $T_q$, at a constant value of 1.0. We ought to emphasize the significance of maintaining this uniform temperature setting as it plays a pivotal role in speculative decoding, by ensuring a consistent and coherent semantic interpretation for both $\mathcal{M}_p$ and $\mathcal{M}_q^\theta$. A summary of the hyperparameters used in our knowledge distillation process can be found in Table B.1.

Table B.1: Hyperparameters for distillation experiments.

| hyperparameter | value |
|---|---|
| training steps | 300,000 |
| batch size | 32 |
| dropout | 0.0 |
| learning rate (LR) | 0.0003 |
| LR warmup steps | 5,000 |
| LR cooldown (begin, end) | (150,000, 300,000) |
| warmup schedule | linear (from 0 to LR) |
| cooldown schedule | cosine decay (from LR to 0.1LR) |

### B.4 EVALUATION

To obtain scores for each task (specifically, ROUGE-2 for XSum and CNN/DM, BLEU for WMT, and accuracy for GSM8K), we employ the evaluation methodology in Agarwal et al. (2023). We assess models on all examples in the test or validation sets and report their average performance. To assess empirical speculative decoding performance, i.e., the empirical acceptance rate and empirical block efficiency, we conduct evaluations on all instances in the test or validation sets and report the average value of these metrics.

To measure the actual latency, we follow Leviathan et al. (2023) and execute both our target model and draft models are on the same TPUv4 device without utilizing model parallelism. We randomly sample 500 examples from either the test or validation set, and measure the wall-clock decoding time on a batch size of 1. This measurement procedure is repeated three times, and the mean performance is reported. We have observed minimal variance across different random seeds in our results.

# C ADDITIONAL RESULTS

## C.1 ENHANCING SPECULATIVE DECODING THROUGH KNOWLEDGE DISTILLATION

Table C.1: DistillSpec improves speculative decoding performance across various datasets and block sizes, for both greedy decoding and temperature sampling. BBH-AVG contains the evaluation of the distilled draft model from GSM8K on all BIG-Bench Hard (BBH) tasks (Suzgun et al., 2022); we report the average results over the 23 BBH tasks. See Figure C.2 and C.3 for a detailed breakdown of performance on individual BBH tasks.

| Dataset | Model | Sampling | | w/o Distillation | | with Distillation | | | |
|---|---|---|---|---|---|---|---|---|---|
| | | temp | $\gamma$ | $\tau$ | SPEED | $\tau$ | SPEED | $\Delta$ | Method |
| XSum | Target $M_p$ T5-XL (3B) | T=0 | 3 | 2.31 | 1.44$\times$ | 2.62 | 1.58$\times$ | +0.14$\times$ | $f$-Distill |
| | | | 5 | 2.57 | 1.43$\times$ | 3.08 | 1.62$\times$ | +0.19$\times$ | $f$-Distill |
| | | | 7 | 2.68 | 1.36$\times$ | 3.31 | 1.57$\times$ | +0.21$\times$ | $f$-Distill |
| | Draft $M_q$ T5-Small (77M) | T=1 | 3 | 2.19 | 1.40$\times$ | 2.58 | 1.57$\times$ | +0.17$\times$ | $f$-Distill |
| | | | 5 | 2.39 | 1.37$\times$ | 3.01 | 1.61$\times$ | +0.25$\times$ | $f$-Distill |
| | | | 7 | 2.47 | 1.28$\times$ | 3.21 | 1.55$\times$ | +0.27$\times$ | $f$-Distill |
| CNNDM | Target $M_p$ T5-XL (3B) | T=0 | 3 | 2.83 | 1.89$\times$ | 3.19 | 2.11$\times$ | +0.22$\times$ | $f$-Distill |
| | | | 5 | 3.46 | 2.07$\times$ | 4.13 | 2.42$\times$ | +0.35$\times$ | GKD |
| | | | 7 | 3.85 | 2.07$\times$ | 4.83 | 2.53$\times$ | +0.46$\times$ | $f$-Distill |
| | Draft $M_q$ T5-Small (77M) | T=1 | 3 | 2.49 | 1.71$\times$ | 2.87 | 1.93$\times$ | +0.23$\times$ | $f$-Distill |
| | | | 5 | 2.89 | 1.77$\times$ | 3.52 | 2.12$\times$ | +0.35$\times$ | $f$-Distill |
| | | | 7 | 3.08 | 1.71$\times$ | 3.92 | 2.12$\times$ | +0.41$\times$ | $f$-Distill |
| WMT | Target $M_p$ T5-XL (3B) | T=0 | 3 | 3.22 | 2.08$\times$ | 3.30 | 2.13$\times$ | +0.05$\times$ | $f$-Distill |
| | | | 5 | 4.17 | 2.33$\times$ | 4.32 | 2.41$\times$ | +0.08$\times$ | $f$-Distill |
| | | | 7 | 4.83 | 2.36$\times$ | 5.06 | 2.46$\times$ | +0.10$\times$ | $f$-Distill |
| | Draft $M_q$ T5-Small (77M) | T=1 | 3 | 2.73 | 1.77$\times$ | 2.88 | 1.83$\times$ | +0.07$\times$ | GKD |
| | | | 5 | 3.29 | 1.83$\times$ | 3.48 | 1.93$\times$ | +0.10$\times$ | GKD |
| | | | 7 | 3.54 | 1.72$\times$ | 3.85 | 1.84$\times$ | +0.12$\times$ | $f$-Distill |
| GSM8K | Target $M_p$ T5-XL (3B) | T=0 | 3 | 2.60 | 1.51$\times$ | 2.96 | 1.69$\times$ | +0.18$\times$ | GKD |
| | | | 5 | 3.06 | 1.42$\times$ | 3.68 | 1.65$\times$ | +0.22$\times$ | GKD |
| | | | 7 | 3.27 | 1.36$\times$ | 4.14 | 1.60$\times$ | +0.24$\times$ | GKD |
| | Draft $M_q$ T5-Small (77M) | T=1 | 3 | 2.58 | 1.48$\times$ | 2.84 | 1.64$\times$ | +0.16$\times$ | $f$-Distill |
| | | | 5 | 3.03 | 1.39$\times$ | 3.45 | 1.58$\times$ | +0.19$\times$ | $f$-Distill |
| | | | 7 | 3.23 | 1.33$\times$ | 3.84 | 1.53$\times$ | +0.20$\times$ | $f$-Distill |
| LM1B | Target $M_p$ GPT-Like (234M) | T=0 | 3 | 2.96 | 3.66$\times$ | 3.13 | 3.97$\times$ | +0.31$\times$ | $f$-Distill |
| | | | 5 | 3.69 | 3.35$\times$ | 3.92 | 3.51$\times$ | +0.16$\times$ | $f$-Distill |
| | | | 7 | 4.15 | 2.52$\times$ | 4.55 | 2.72$\times$ | +0.20$\times$ | $f$-Distill |
| | Draft $M_q$ GPT-Like (33M) | T=1 | 3 | 2.51 | 2.34$\times$ | 2.69 | 2.45$\times$ | +0.11$\times$ | $f$-Distill |
| | | | 5 | 2.90 | 2.79$\times$ | 3.20 | 3.02$\times$ | +0.23$\times$ | $f$-Distill |
| | | | 7 | 3.10 | 1.98$\times$ | 3.51 | 2.18$\times$ | +0.20$\times$ | $f$-Distill |
| BBH-AVG | Target $M_p$ T5-XL (3B) | T=0 | 3 | 2.55 | 1.90$\times$ | 2.68 | 2.06$\times$ | +0.16$\times$ | $f$-Distill |
| | | | 5 | 2.98 | 1.92$\times$ | 3.19 | 2.16$\times$ | +0.24$\times$ | $f$-Distill |
| | | | 7 | 3.20 | 1.93$\times$ | 3.49 | 2.21$\times$ | +0.28$\times$ | $f$-Distill |
| | Draft $M_q$ T5-Small (77M) | T=1 | 3 | 2.45 | 1.79$\times$ | 2.57 | 1.93$\times$ | +0.14$\times$ | $f$-Distill |
| | | | 5 | 2.81 | 1.80$\times$ | 3.03 | 1.98$\times$ | +0.18$\times$ | $f$-Distill |
| | | | 7 | 3.01 | 1.78$\times$ | 3.28 | 2.02$\times$ | +0.23$\times$ | $f$-Distill |

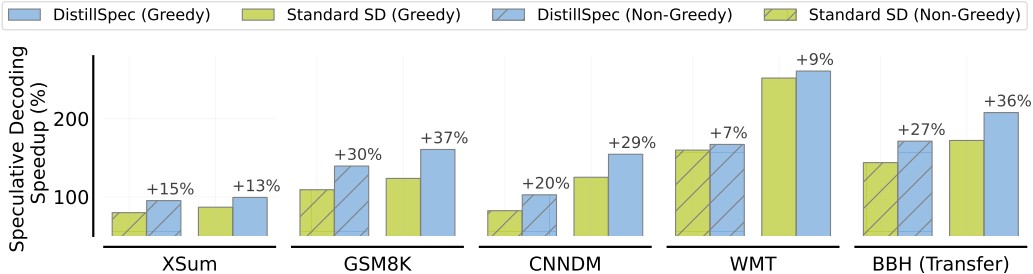

Figure C.1: The distilled draft T5-Small model, derived from a T5-XL teacher model, is capable of generalizing to a larger target model (T5-XXL), resulting in a substantial acceleration in various scenarios. BBH-AVG contains the evaluation of the distilled draft model from GSM8K on all BIG-Bench Hard (BBH) tasks (Suzgun et al., 2022); we report the average results over the 23 BBH tasks. See Figure C.4 and C.5 for a detailed breakdown of performance on individual BBH tasks.

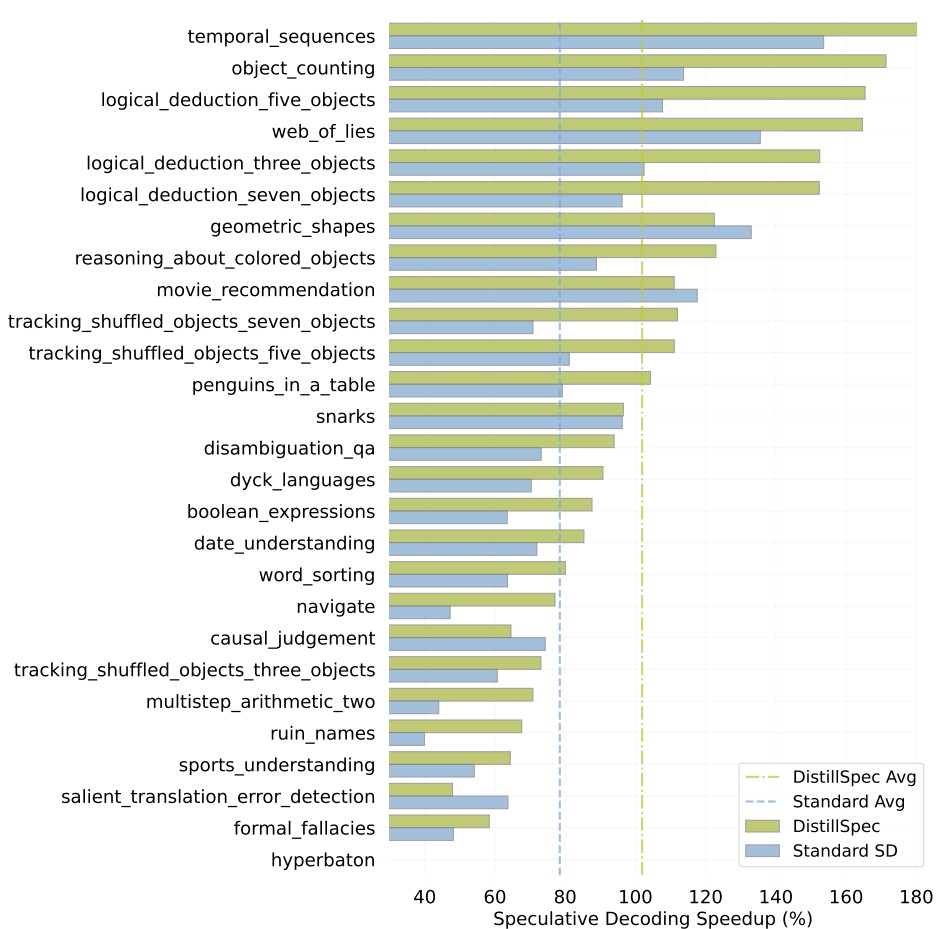

Figure C.2: Assessing DistillSpec's model transferability on the BIG-Bench Hard suite: zero-shot CoT reasoning with non-greedy decoding. This study examines a T5-Small draft model, initially trained on the GSM8K dataset, across 23 varied tasks using T5-XL as the target model. DistillSpec can deliver significant speculative decoding speedups on a broad spectrum of tasks.

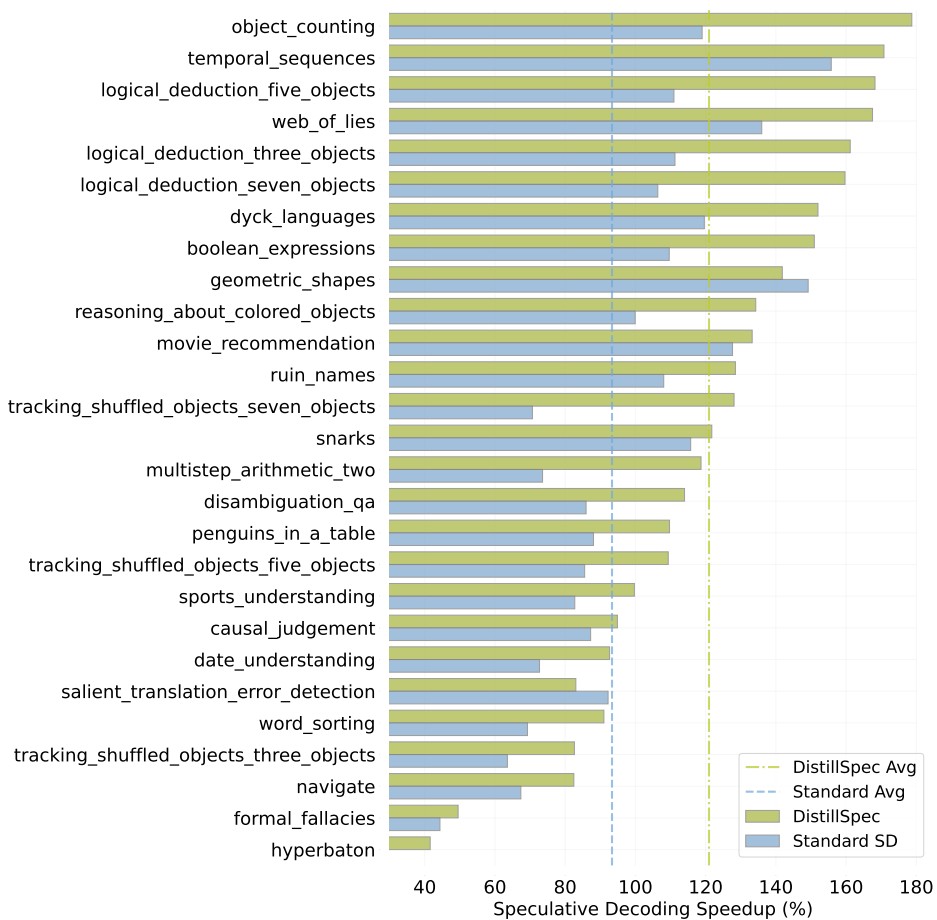

Figure C.3: Assessing DistillSpec's model transferability on the BIG-Bench Hard suite: zero-shot CoT reasoning with greedy decoding. This study examines a T5-Small draft model, initially trained on the GSM8K dataset, across 23 varied tasks using T5-XL as the target model. DistillSpec can deliver significant speculative decoding speedups on a broad spectrum of tasks.

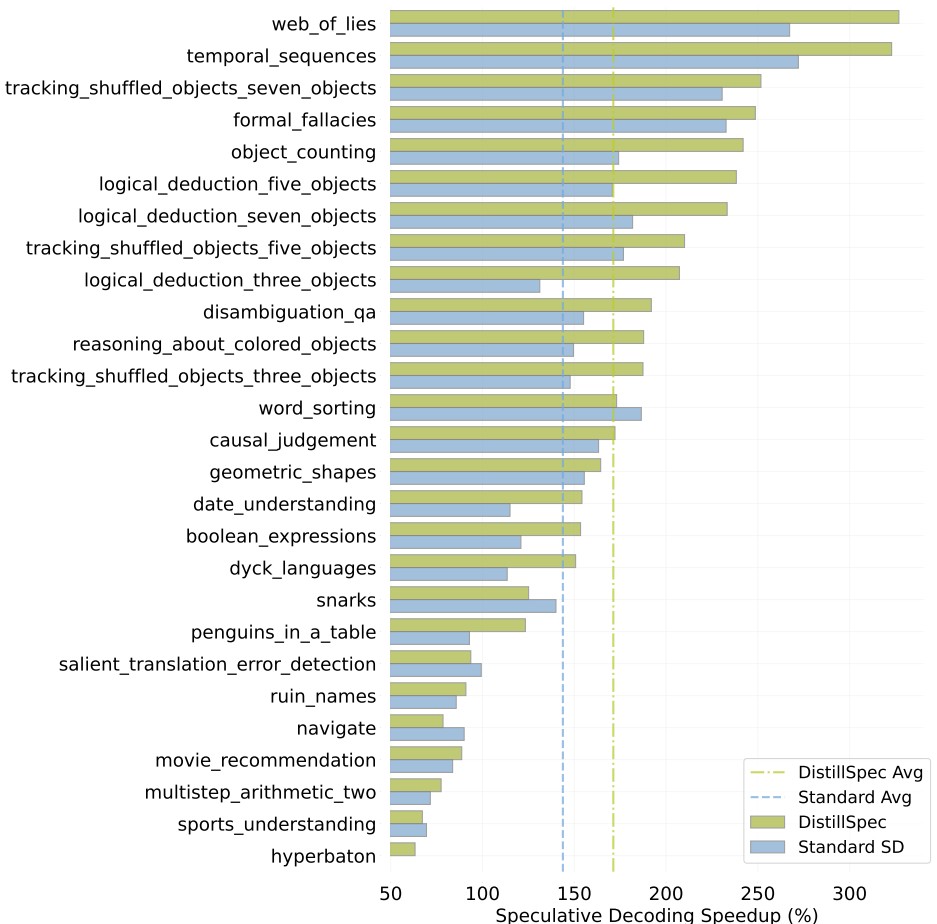

Figure C.4: Assessing DistillSpec's model transferability on the BIG-Bench Hard suite: zero-shot CoT reasoning with non-greedy decoding. This study examines a T5-Small draft model, initially trained on the GSM8K dataset, across 23 varied tasks using T5-XXL as the target model. DistillSpec can deliver significant speculative decoding speedups on a broad spectrum of tasks.

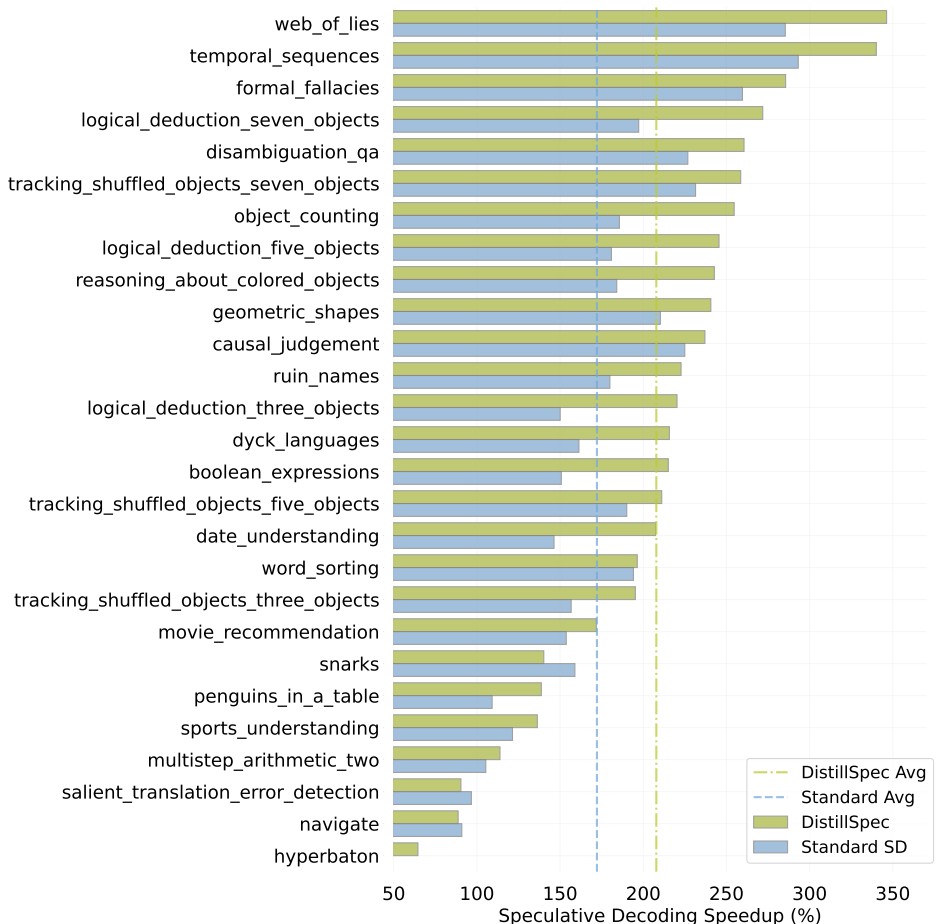

Figure C.5: Assessing DistillSpec's model transferability on the BIG-Bench Hard suite: zero-shot CoT reasoning with greedy decoding. This study examines a T5-Small draft model, initially trained on the GSM8K dataset, across 23 varied tasks using T5-XXL as the target model. DistillSpec can deliver significant speculative decoding speedups on a broad spectrum of tasks.

### C.1.1 EMPIRICAL BLOCK EFFICIENCY IMPROVEMENT

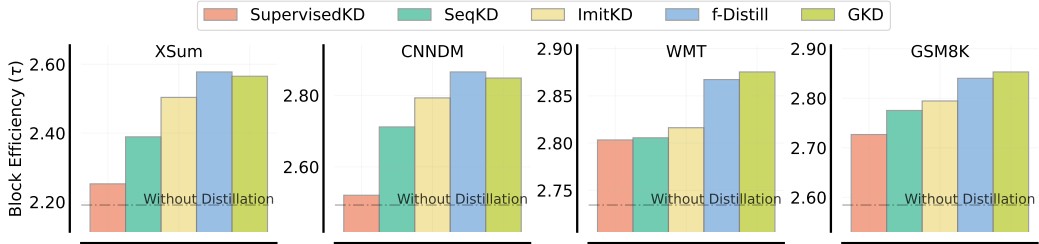

Figure C.6: Empirical block efficiency improvement of DistillSpec for non-greedy sampling ($T = 1$) and block size $\gamma = 3$. The draft model is trained using one of the distillation methods listed in Table 1 of Section 4. The dashed line indicates the block efficiency of speculative decoding using a non-distilled draft model. DistillSpec outperforms standard speculative decoding across all of the distillation methods being considered, with f-Distill and GKD yielding the highest gains.

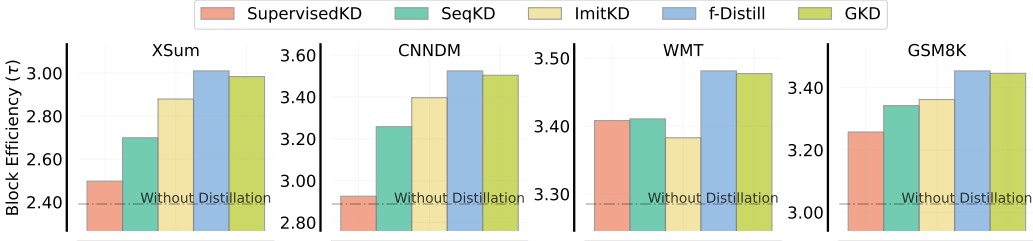

Figure C.7: Empirical block efficiency improvement of DistillSpec for non-greedy sampling ($T = 1$) and block size $\gamma = 5$. The draft model is trained using one of the distillation methods listed in Table 1 of Section 4. The dashed line indicates the block efficiency of speculative decoding using a non-distilled draft model. DistillSpec outperforms standard speculative decoding across all of the distillation methods being considered, with f-Distill and GKD yielding the highest gains.

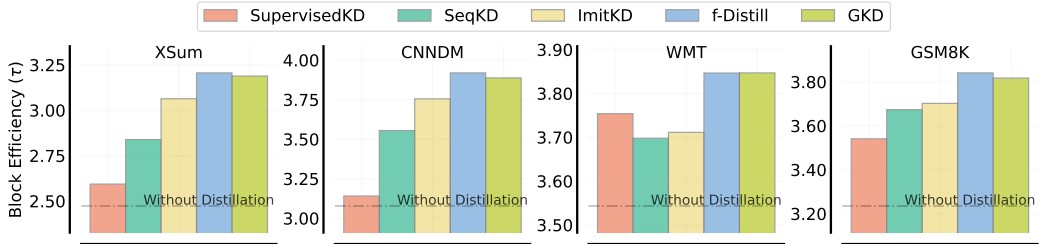

Figure C.8: Empirical block efficiency improvement of DistillSpec for non-greedy sampling ($T = 1$) and block size $\gamma = 7$. The draft model is trained using one of the distillation methods listed in Table 1 of Section 4. The dashed line indicates the block efficiency of speculative decoding using a non-distilled draft model. DistillSpec outperforms standard speculative decoding across all of the distillation methods being considered, with f-Distill and GKD yielding the highest gains.

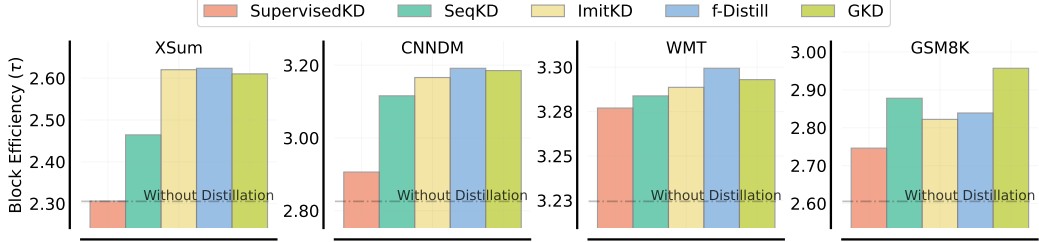

Figure C.9: Empirical block efficiency improvement of DistillSpec for greedy sampling ($T = 0$) and block size $\gamma = 3$. The draft model is trained using one of the distillation methods listed in Table 1 of Section 4. The dashed line indicates the block efficiency of speculative decoding using a non-distilled draft model. DistillSpec outperforms standard speculative decoding across all of the distillation methods being considered, with GKD weakly outperforming the other methods on average.

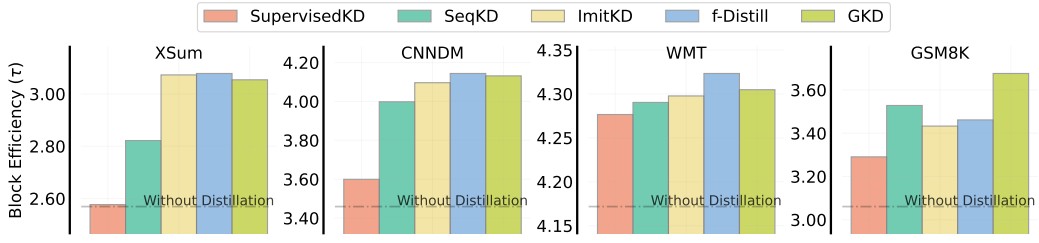

Figure C.10: Empirical block efficiency improvement of DistillSpec for greedy sampling ($T = 0$) and block size $\gamma = 5$. The draft model is trained using one of the distillation methods listed in Table 1 of Section 4. The dashed line indicates the block efficiency of speculative decoding using a non-distilled draft model. DistillSpec outperforms standard speculative decoding across all of the distillation methods being considered, with GKD weakly outperforming the other methods on average.

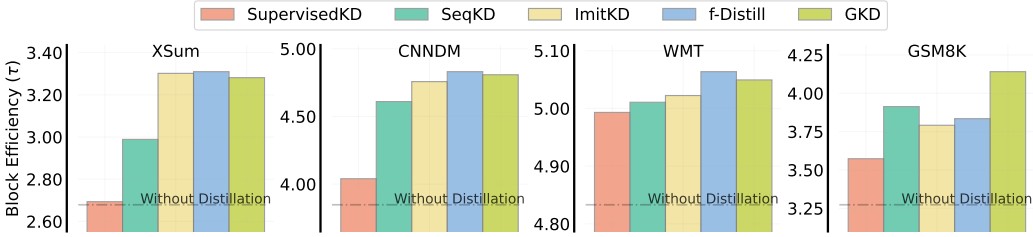

Figure C.11: Empirical block efficiency improvement of DistillSpec for greedy sampling ($T = 0$) and block size $\gamma = 7$. The draft model is trained using one of the distillation methods listed in Table 1 of Section 4. The dashed line indicates the block efficiency of speculative decoding using a non-distilled draft model. DistillSpec outperforms standard speculative decoding across all of the distillation methods being considered, with GKD weakly outperforming the other methods on average.

## C.1.2 PERFORMANCE IMPROVEMENT OVER TIME

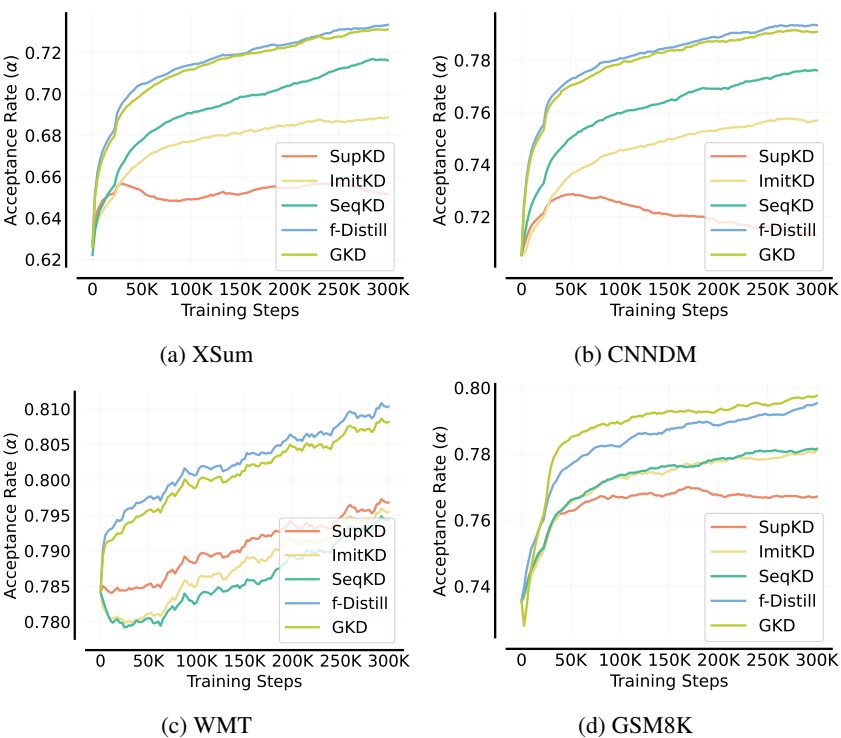

(a) XSum

(b) CNNDM

(c) WMT

(d) GSM8K

Figure C.12: Progression of the acceptance rate $\alpha$ of DistillSpec over the training of the draft model, measured by the number of training steps. The draft model is trained using one of the distillation methods listed in Table 1 of Section 4. GKD and $f$-Distill yields the most consistent improvement in $\alpha$ over training steps, while SupKD yields the least improvement and exhibits declining acceptance rates after ~40k training steps on XSum and CNNDM.

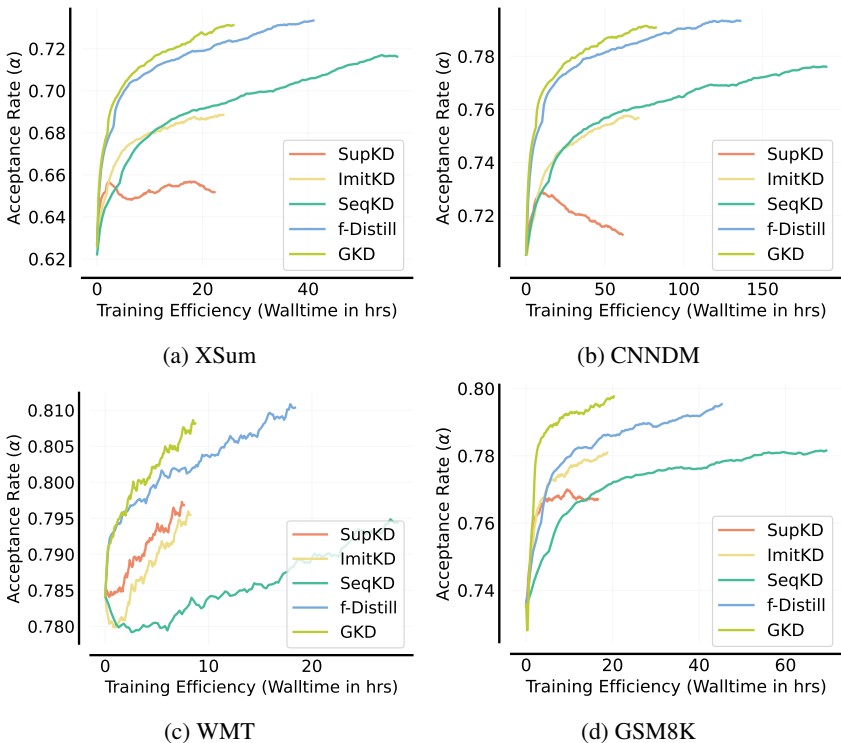

(a) XSum

(b) CNNDM

(c) WMT

(d) GSM8K

Figure C.13: Progression of the acceptance rate of DistillSpec over the training of the draft model, measured by the training wall time. The draft model is trained using one of the distillation methods listed in Table 1 of Section 4. GKD and $f$-Distill yield the most consistent improvement in $\alpha$ over training wall time, while SupKD yields the least improvement and even exhibits an inflection in the acceptance rate early during training.

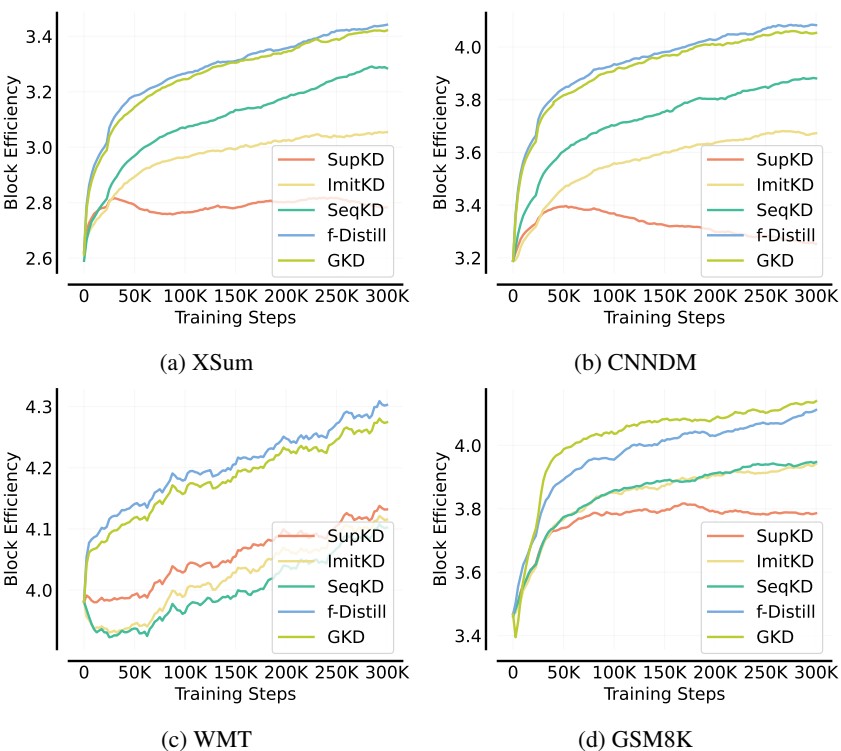

Figure C.14: Progression of the block efficiency $\tau$ of DistillSpec over the training of the draft model, measured by the number of training steps. The draft model is trained using one of the distillation methods listed in Table 1 of Section 4. GKD and $f$-Distill yield the most consistent improvement in $\tau$ over training, while SupKD yields the least improvement and exhibits a drop in block efficiency after $\sim$40k training steps on XSum and CNNDM.

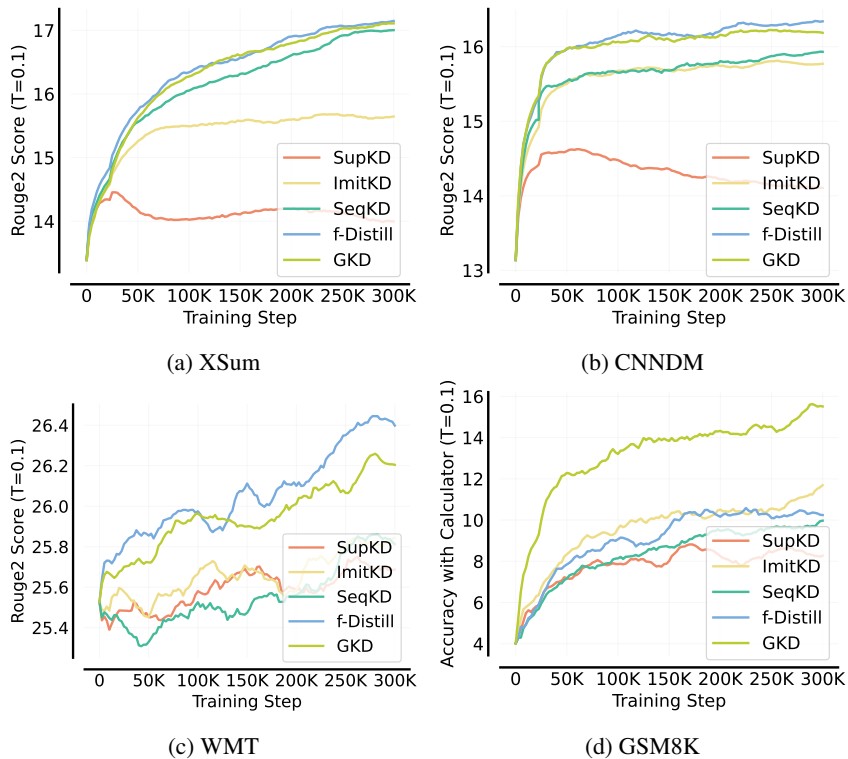

Figure C.15: Progression of the task performance of the draft model over the course of its training, as measured by the number of training steps. Task performance is based on sampling temperature $T = 0.1$ and is measured by the Rouge2 score on the XSum and CNNDM datasets, and the accuracy score with the help of a calculator on GSM8K. The draft model is trained using one of the distillation methods listed in Table 1 of Section 4. GKD and f-Distill yield the most consistent improvement in task performance over training, while SupKD yields the least improvement and exhibits declining Rouge2 scores after ~40k training steps on XSum and CNNDM.

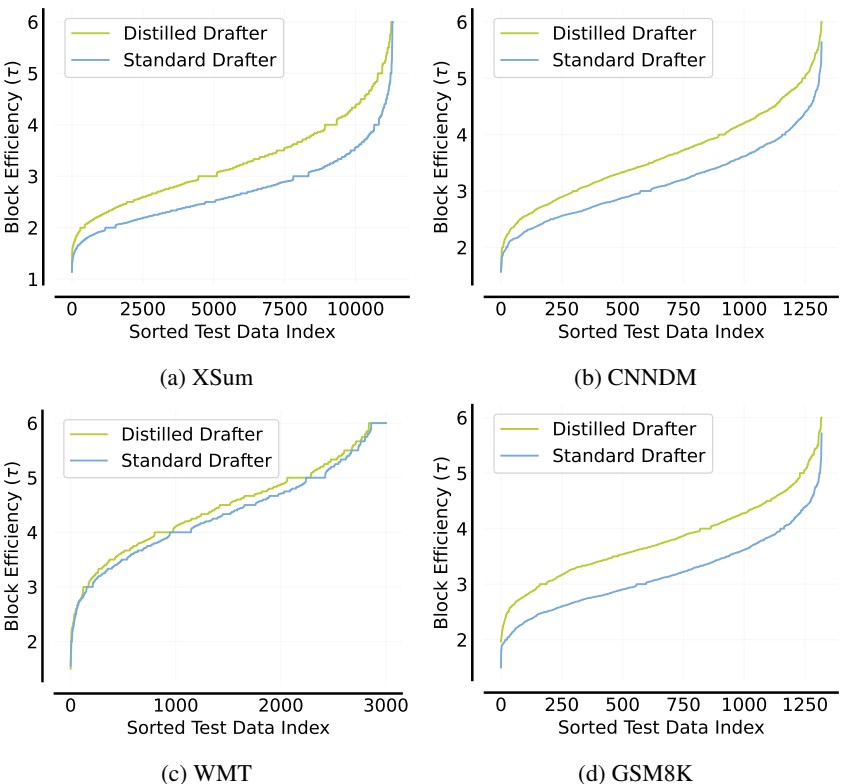

Figure C.16: Comparison of the block efficiency of draft models trained with distillation vs. without distillation. The draft model is trained using f-Distill on XSum and CNNDM, and on-policy GKD on GSM8K. For each dataset and draft model, the block efficiency is sorted and plotted from lowest to highest index. The higher position of the curve for the distilled draft model at each index indicates that the distribution of block efficiency of the distilled draft model *first-order stochastically dominates* the block efficiency of the non-distilled draft model.

### C.1.3 SAMPLING TEMPERATURE EFFECT

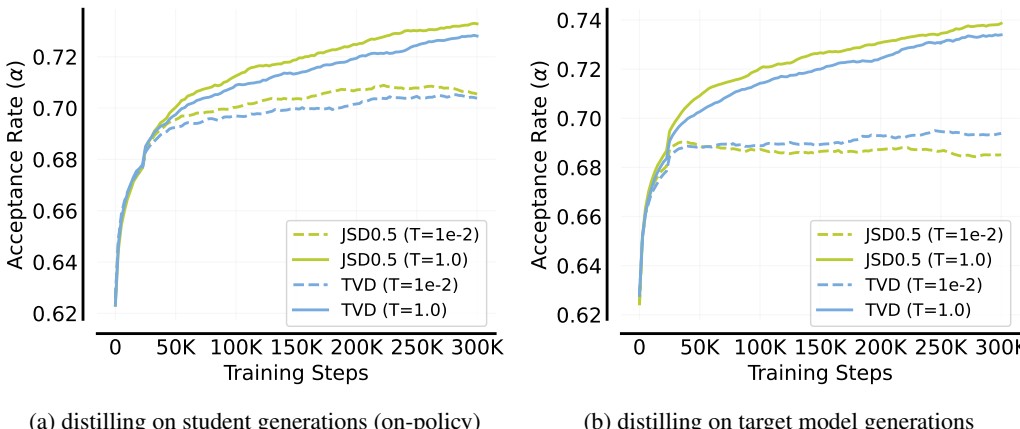

(a) distilling on student generations (on-policy)    (b) distilling on target model generations

Figure C.17: Effect of the sampling temperature on the acceptance rate in XSum decoding. We assess how varying the sampling temperature influences speculative decoding performance in two distinct scenarios: (a) the draft model is trained on its own generated sequences (on-policy distillation), and (b) the draft model is trained on sequences sampled from the target model. In both instances, we find that using a high sampling temperature is paramount for attaining superior performance.

### C.1.4 REDUCTION IN CROSS-ENTROPY

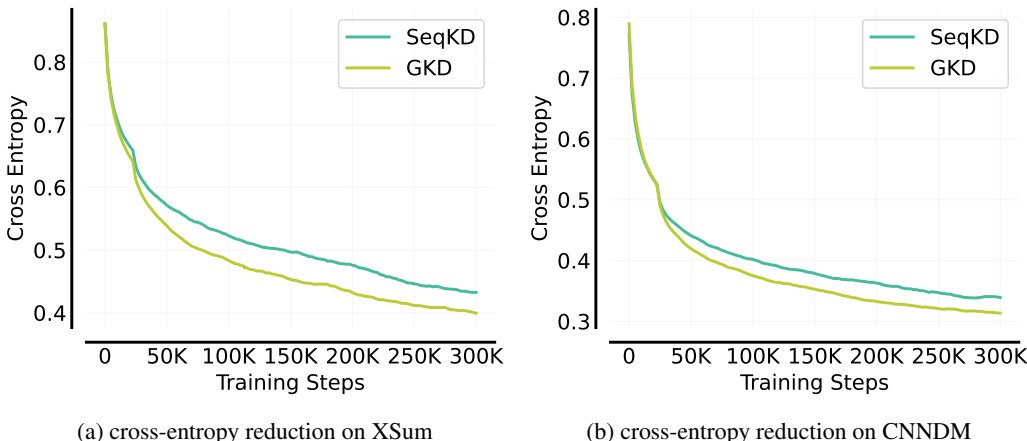

(a) cross-entropy reduction on XSum    (b) cross-entropy reduction on CNNDM

Figure C.18: BiLD (Kim et al., 2023) proposes an SD variant that uses the cross-entropy between the target and draft model as the rejection criterion and uses SeqKD to improve the alignment. Our experiment shows that GKD can reduce the cross-entropy more effectively than SeqKD. Thus, DistillSpec can potentially be utilized to improve BiLD.

## C.2 DISTILLATION RECIPE

### C.2.1 SCORE AND BLOCK EFFICIENCY IMPROVEMENT

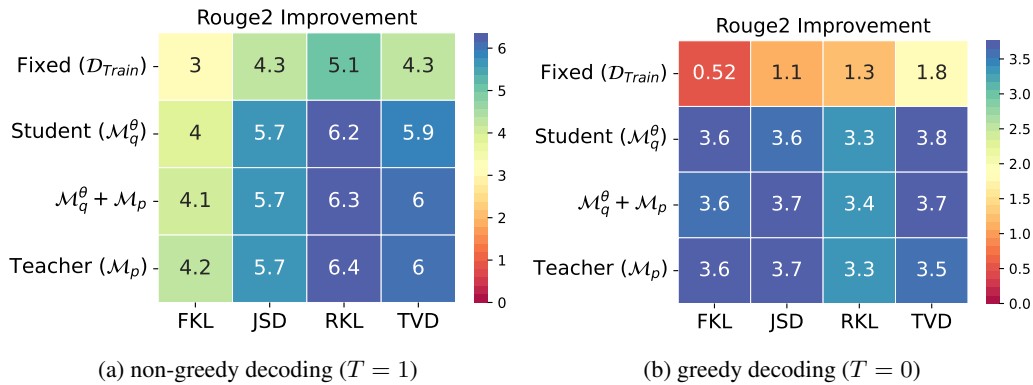

(a) non-greedy decoding ($T = 1$)  (b) greedy decoding ($T = 0$)

Figure C.19: Improvement in the Rouge2 score of DistillSpec over standard speculative decoding on XSum given by different combinations of divergence functions and generative distributions used in the draft model distillation (see Table 1 of Section 4), using greedy ($T = 0$) and non-greedy ($T = 1$) sampling.

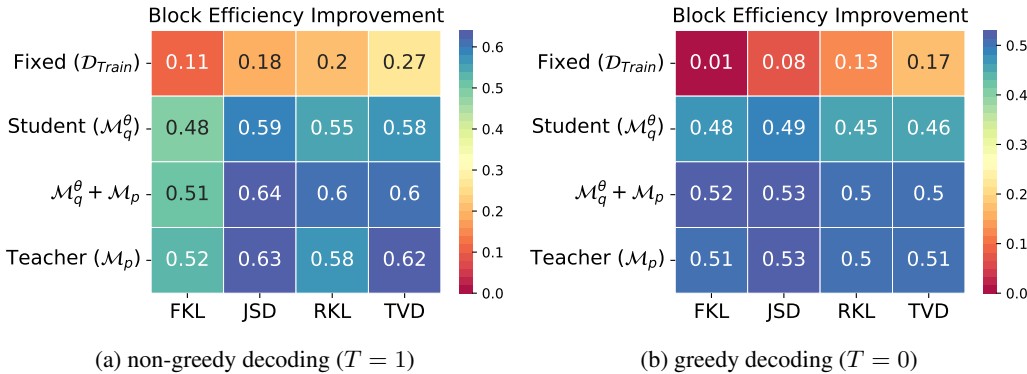

(a) non-greedy decoding ($T = 1$)  (b) greedy decoding ($T = 0$)

Figure C.20: Improvement in the block efficiency of DistillSpec over standard speculative decoding on XSum given by different combinations of divergence functions and generative distributions used in the draft model distillation (see Table 1 of Section 4), using greedy ($T = 0$) and non-greedy ($T = 1$) sampling.

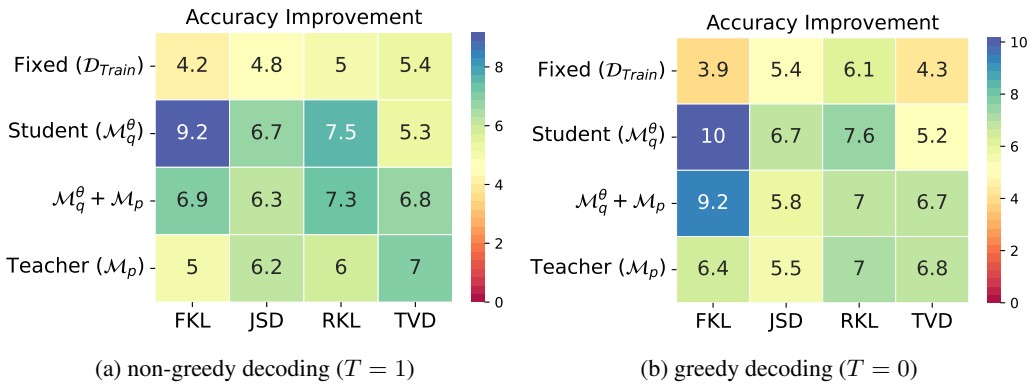

Figure C.21: Improvement in the accuracy score of DistillSpec over standard speculative decoding on GSM8K given by different combinations of divergence functions and generative distributions used in the draft model distillation (see Table 1 of Section 4), using greedy ($T = 0$) and non-greedy ($T = 1$) sampling.

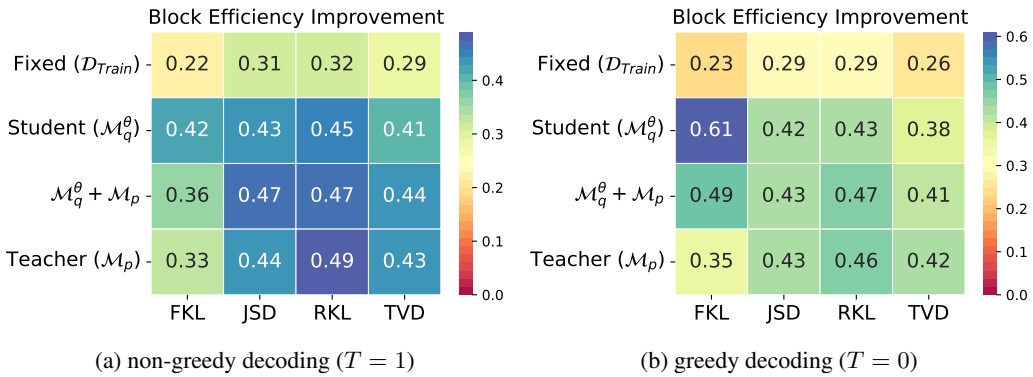

Figure C.22: Improvement in the block efficiency of DistillSpec over standard speculative decoding on GSM8K given by different combinations of divergence functions and generative distributions used in the draft model distillation (see Table 1 of Section 4), using greedy ($T = 0$) and non-greedy ($T = 1$) sampling.

### C.2.2 IMPACT OF DISTILLATION ON DRAFT QUALITY VS. COMPATIBILITY

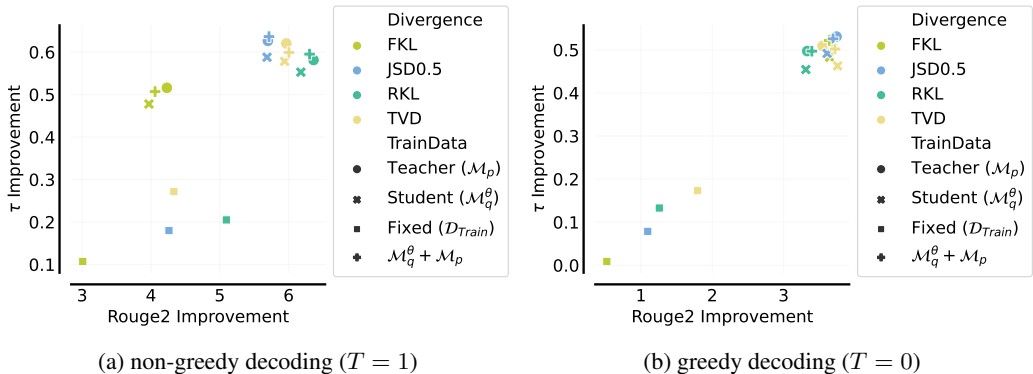

    (a) non-greedy decoding ($T = 1$)          (b) greedy decoding ($T = 0$)

Figure C.23: Correlation plot of the improvements in the block efficiency (y-axis) and Rouge2 score (x-axis) of DistillSpec over standard speculative decoding on XSum given by different combinations of divergence functions and generative distributions used in the draft model distillation (see Table 1 of Section 4), using greedy ($T = 0$) and non-greedy ($T = 1$) sampling.

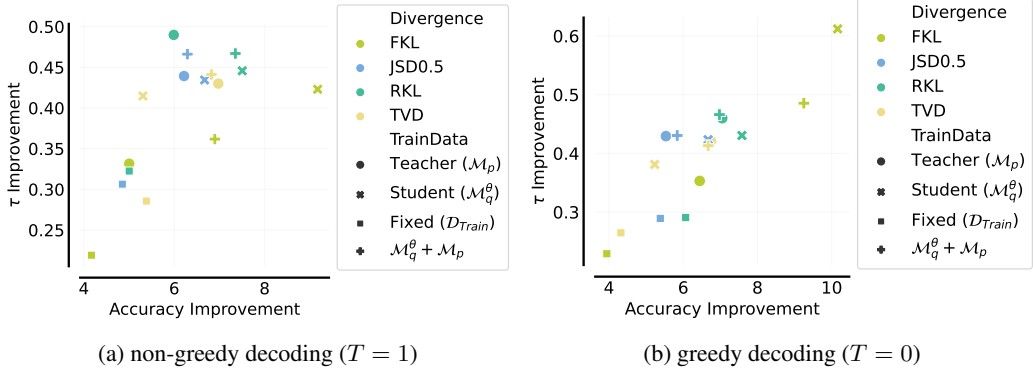

    (a) non-greedy decoding ($T = 1$)          (b) greedy decoding ($T = 0$)

Figure C.24: Correlation plot of the improvements in the block efficiency (y-axis) and Accuracy (x-axis) of DistillSpec over standard speculative decoding on GSM8K given by different combinations of divergence functions and generative distributions used in the draft model distillation (see Table 1 of Section 4), using greedy ($T = 0$) and non-greedy ($T = 1$) sampling.

### C.3 Quality versus latency trade-off

#### C.3.1 Lossy speculative decoding

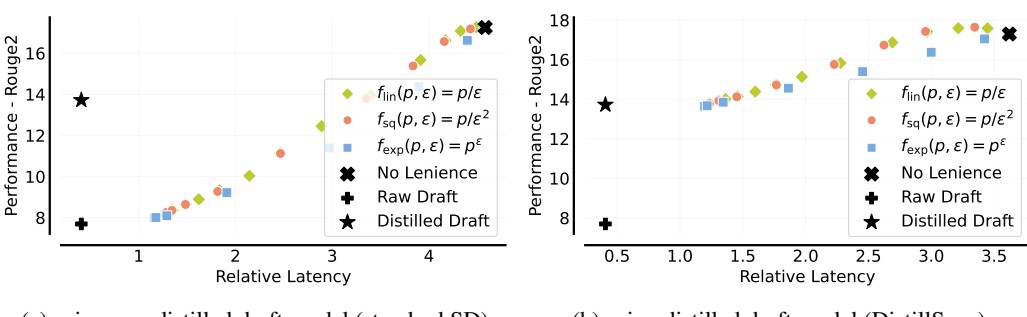

(a) using non-distilled draft model (standard SD)  (b) using distilled draft model (DistillSpec)

Figure C.25: Tradeoff between task performance and decoding latency on XSum enabled by combining lossy speculative decoding with the alternative variants of DistillSpec presented in Section 4. The higher position and flatter slope of the tradeoff curve for DistillSpec indicate that the method enables larger latency gains for smaller reduction in model quality than are feasible under standard SD.

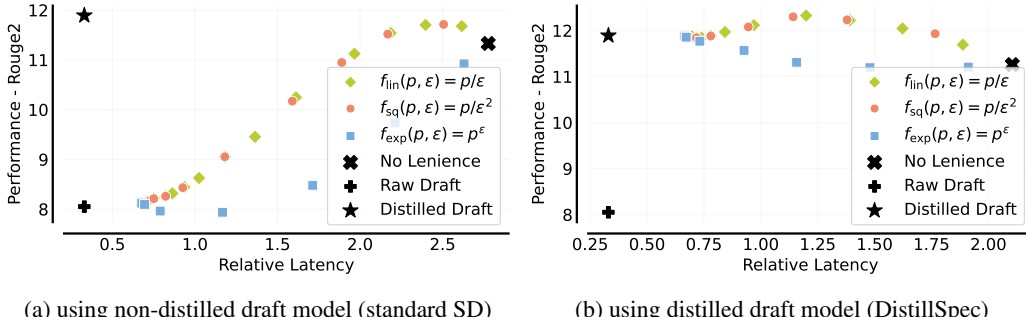

(a) using non-distilled draft model (standard SD)  (b) using distilled draft model (DistillSpec)

Figure C.26: Tradeoff between task performance and decoding latency on CNNDM enabled by combining lossy speculative decoding with the alternative variants of DistillSpec presented in Section 4. The higher position and flatter slope of the tradeoff curve for DistillSpec indicate that the method enables larger latency gains for smaller reduction in model quality than are feasible under standard SD.

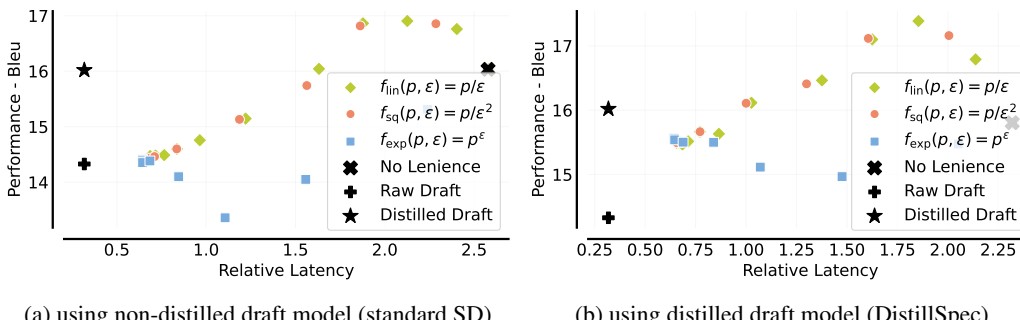

(a) using non-distilled draft model (standard SD)          (b) using distilled draft model (DistillSpec)

Figure C.27: Tradeoff between task performance and decoding latency on WMT enabled by combining lossy speculative decoding with the alternative variants of DistillSpec presented in Section 4. While the similar height and slope of the tradeoff curves of DistillSpec and standard SD indicate a comparable quality-latency tradeoff between the two setups, DistillSpec exhibits lower latency overall.

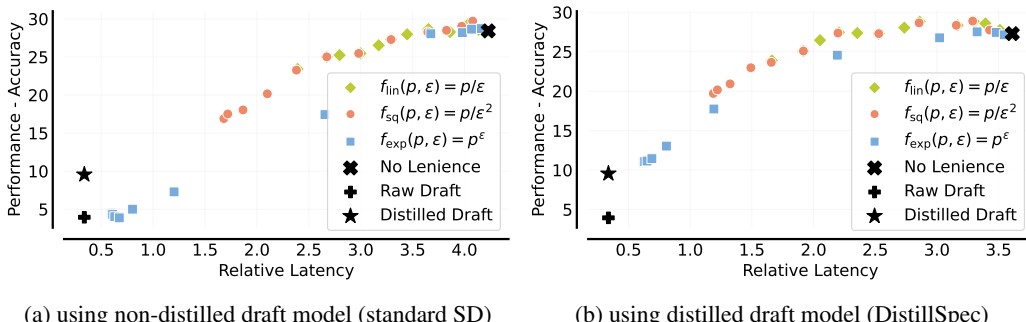

(a) using non-distilled draft model (standard SD)          (b) using distilled draft model (DistillSpec)

Figure C.28: Tradeoff between task performance and decoding latency on GSM8K enabled by combining lossy speculative decoding with the alternative variants of DistillSpec presented in Section 4. While the similar height and slope of the tradeoff curves of DistillSpec and standard SD indicate a comparable quality-latency tradeoff between the two setups, DistillSpec exhibits lower latency overall.

### C.3.2 DISTILLSPEC MEETS MODEL GARDEN

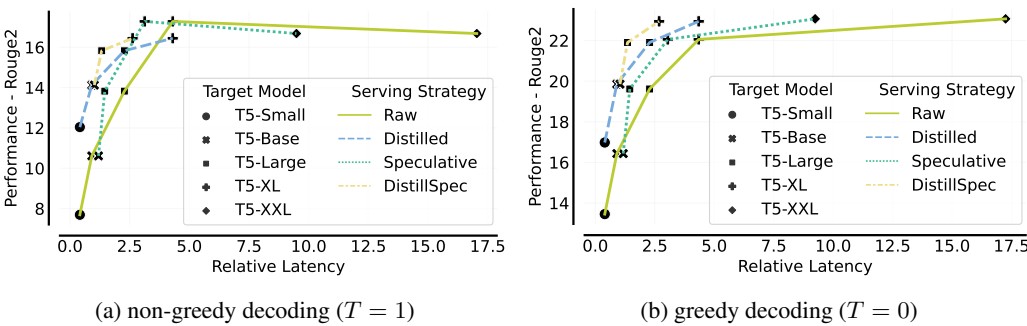

(a) non-greedy decoding ($T = 1$)  (b) greedy decoding ($T = 0$)

Figure C.29: Quality-latency trade-off curves on XSum using greedy ($T = 0$) or non-greedy ($T = 1$) sampling, for a single target LLM trained without distillation ("Raw"), trained by distilling from a larger LLM ("Distilled"), speculative decoding using a draft model trained without distillation ("Speculative"), and speculative decoding combining a distilled target LLM with a distilled draft model ("DistillSpec").

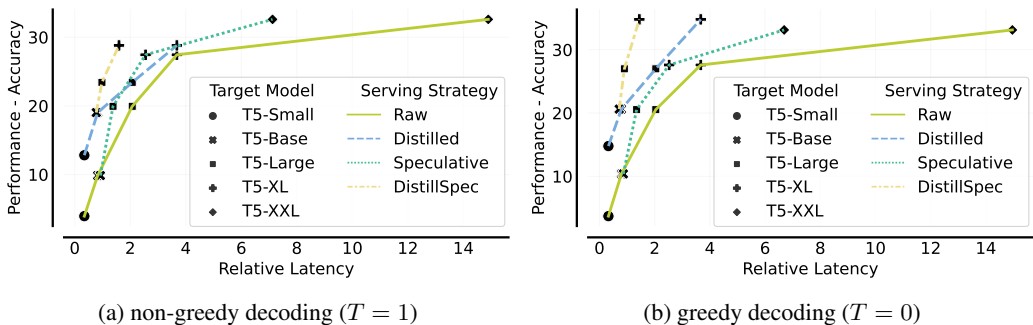

(a) non-greedy decoding ($T = 1$)  (b) greedy decoding ($T = 0$)

Figure C.30: Quality-latency trade-off curves on GSM8K using greedy ($T = 0$) or non-greedy ($T = 1$) sampling, for a single target LLM trained without distillation ("Raw"), trained by distilling from a larger LLM ("Distilled"), speculative decoding using a draft model trained without distillation ("Speculative"), and speculative decoding using a distilled target LLM combining a distilled draft model ("DistillSpec").

