# OpenReview forum: "DistillSpec: Improving Speculative Decoding via Knowledge Distillation"
_ICLR.cc/2024/Conference — ICLR 2024 poster_

### Official Review · Reviewer_p4vK · 2023-11-01

**Soundness:** 3 good
**Presentation:** 3 good
**Contribution:** 3 good
**Rating:** 6
**Confidence:** 4

**Summary:**

The authors propose DistillSpec that uses knowledge distillation to better align the draft model with the target model. The authors have a wide exploration of different dilation methods from SeqKD to on-plicy GKD. DistillSpec yields impressive 10 − 45% speedups over standard speculative decoding on a range of standard benchmarks.

**Strengths:**

1. The experiments are quite solid. It explores different distillation losses and different ways to collect data for distillation. All the tables and figures are well written, and it is easy to get the difference between different methods.
2. The authors also explore the trade-off between accuracy and latency.
3. The paper is well-written and easy to read.

**Weaknesses:**

1. All the distillation methods are based on existing works. The work is more of a comprehensive study of knowledge distillation on LLM for speculative decoding.
2. All the experiments are done in an in-domain setting, while the benefits of LLM are on zeroshot/fewshot setting on out-of-domain datasets. Thus the experiments can not show whether it can replace the general draft models.

**Questions:**

Why do different distillation methods vary a lot in Figure 4?

---

> ### Author Response · Authors · 2023-11-19
> **Author Response: Clarify the novelty and contributions. Clarify the performance variance. Zero-shot CoT evaluation on BigBenchHard (23 tasks).**
>
> We thank the reviewer for taking the time to provide your thoughtful and valuable feedback on our work. Below are responses to your concerns.
>
> > *All the distillation methods are based on existing works. The work is more of a comprehensive study of knowledge distillation on LLM for speculative decoding.*
>
> Indeed, **the goal of our paper is not to propose a new KD algorithm but to provide a way to improve SD using KD**. That said, applying KD to SD is not well-explored in the past literature. As shown in Figure 2, 3a, the previous attempt [1] using SeqKD turned out to be not very effective to improve SD. Therefore, we provide a systematic and comprehensive study on the most cost effective way to apply KD to SD.
>
> To understand the key factors that affect the distillation performance, we categorize most commonly used KD techniques in terms of “divergence function” and “training data'' in Table 1 and investigate many other KD variants in Section 5.2 (Figure 4). Note that many combinations of the divergence function and training data have not been studied in the past literature. We believe that our findings that 1) using student generated data is most cost-effective; and 2) the best divergence function depends on tasks and decoding strategy; are meaningful and provide new insights on how we should apply distillation in practice to improve LLM latency.
>
> Furthermore, we wish to emphasize our contributions to the study of the quality and latency trade-off, a central challenge in LLM inference. Our research explores this trade-off through two innovative perspectives: 'lossy speculative decoding' and 'model garden'. The former offers fine-grained control over the trade-off curve, while the latter presents optimal strategies for combining SD and Distillation. These contributions are novel and highly pertinent for practitioners seeking to balance quality and latency in model deployment.
>
> [1] Stern, M., Shazeer, N., & Uszkoreit, J. (2018). Blockwise parallel decoding for deep autoregressive models. Advances in Neural Information Processing Systems, 31.
>
> > *All the experiments are done in an in-domain setting, while the benefits of LLM are on zeroshot/fewshot settings on out-of-domain datasets. Thus the experiments can not show whether it can replace the general draft models.*
>
> Thanks for the suggestion! We have added a new experiment to study the transferability of the distilled draft model across different datasets. Specifically, we take the distilled draft model from GSM8K and evaluate it on all **BigBenchHard** tasks (23 tasks in total and all are out-of-domain) [1] with **zero-shot CoT**. We observe that the distilled draft model transfers well to other datasets. Compared to standard SD, it improves the average decoding speed from 1.93x and 1.78x to 2.21x and 2.02x using the greedy and non-greedy decoding, respectively. See Appendix Figure C.2 and C.3 for a detailed breakdown of the tasks in BBH.
>
> [1] Suzgun, M., Scales, N., Schärli, N., Gehrmann, S., Tay, Y., Chung, H. W., ... & Wei, J. (2022). Challenging big-bench tasks and whether chain-of-thought can solve them. arXiv preprint arXiv:2210.09261.
>
> > *Why do different distillation methods vary a lot in Figure 4?*
>
> Indeed, it is an interesting finding from our comprehensive study on divergence function in Section 5.2. Empirically, we observe that different divergence functions perform differently on different tasks. It may be due to certain specific characteristics of the tasks but we do not have a good explanation yet. While this performance variance is disappointing, we think it truthfully reflects the fact that distillation in the real world is challenging. Therefore, we recommend to view the divergence function as one of the hyperparameters in a distillation algorithm and we should tune it just like any other hyperparameters.
>
> -----
> *We hope that most of the reviewer’s concerns have been addressed and, if so, we would appreciate it if they could reconsider their assessment. We’d be happy to engage in further discussions.*

---

### Official Review · Reviewer_GLMj · 2023-11-01

**Soundness:** 3 good
**Presentation:** 2 fair
**Contribution:** 3 good
**Rating:** 6
**Confidence:** 3

**Summary:**

This paper proposes DistillSpec, a method that uses knowledge distillation to improve the speed and quality of speculative decoding. The paper explores various factors that affect the alignment between the draft and target models, such as the training data, the divergence function, and the decoding strategy. It also extends DistillSpec to lossy speculative decoding, which allows for sampling away from the target model distribution. The paper evaluates DistillSpec on several tasks and datasets to demonstrate its speedups.

**Strengths:**

- It provides a comprehensive and systematic study of different aspects of knowledge distillation for speculative decoding, such as data generation, divergence functions, and lossy sampling. The conclusion of models generated data being important makes sense.
- It demonstrates the effectiveness of DistillSpec on several tasks and datasets, using both greedy and non-greedy sampling, and compares it with representative baselines.
- Its lossy speculative decoding results offer novel insights and recommendations for combining knowledge distillation and speculative decoding in different scenarios.

**Weaknesses:**

- The presentation of the paper is a bit messy and unclear. For instance, it is difficult to find the formal definition of DistillSpec among the analysis of various existing distillation approaches; it is not clear how distillation data is generated in detail; the specific configurations of target and draft models in figure 6 are not given, especially for the size of the target model in the DistillSpec case. The clarity and structure of the paper should be improved.
- While DistillSpec is effective on certain tasks, the experiments mainly focus on T5 models and simple tasks (except for GSM8K), compared with the recent advances in LLMs. This also results in the lack of discussion on the difficulty of LLM distillation. Is DistillSpec also effective for LLMs like LLaMA-7B? Can distilling LLMs for some specific tasks also be helpful for other tasks in general? The effectiveness of DistillSpec should be evaluated using more recent large models and some zero-shot benchmarks to prove its effectiveness.
- The paper does not compare or discuss DistillSpec with other methods that combine large and small models at inference, especially under the lossy decoding cases [1].
- The paper does not analyze the generated texts by DistillSpec in greater detail, like the diversity and coherence, which are important aspects of text quality and largely influenced by the sampling approaches. It would be helpful to provide some examples of the generated tasks.

[1] Kim, S., Mangalam, K., Moon, S., Malik, J., Mahoney, M.W., Gholami, A., & Keutzer, K. (2023). Speculative Decoding with Big Little Decoder.

**Questions:**

I am wondering how will DistillSpec be compared with recent work that combines speculative decoding and distillation [1]. It would be interesting to have this discussion.

[1] Liu, X., Hu, L., Bailis, P.D., Stoica, I., Deng, Z., Cheung, A., & Zhang, H. (2023). Online Speculative Decoding. ArXiv, abs/2310.07177.

---

> ### Author Response · Authors · 2023-11-19
> **Author Response: Clarify the definition of DistillSpec, distillation data generation, model architecture choices and model sizes. Zero-shot CoT evaluation on BigBenchHard (23 tasks).**
>
> We thank the reviewer for taking the time to provide your thoughtful and valuable feedback on our work. Below are responses to your concerns.
>
> > *For instance, it is difficult to find the formal definition of DistillSpec among the analysis of various existing distillation approaches*
>
> DistillSpec (Algorithm A.2) refers to our general framework that improves the SD by aligning the target model and draft model using the whitebox knowledge distillation. We do not introduce new divergence functions or new data generation procedures, but instead systematically investigate which combinations of divergence functions and data generation procedures can be best leveraged to improve upon vanilla SD. We have revised the paper to include this definition.
>
> > *it is not clear how distillation data is generated in detail.*
>
> To generate new data, we closely follow the previous work [1]. Specifically, we prompt the model with a task input sampled from the training dataset and ask the model to generate an output. This data generation process works for both teacher and student models.
>
> [1] Agarwal, R., Vieillard, N., Stanczyk, P., Ramos, S., Geist, M., & Bachem, O. (2023). GKD: Generalized Knowledge Distillation for Auto-regressive Sequence Models. arXiv preprint arXiv:2306.13649.
>
> > *the specific configurations of target and draft models in figure 6 are not given, especially for the size of the target model in the DistillSpec case.*
>
> We have updated the figure and summarized the target and draft model size below.
>
> 1) **Raw**: Deploying supervised fine-tuned (SFT) T5 models. It contains five model points: T5-Small (77M), T5-Base (250M), T5-Large (800M), T5-XL (3B), and T5-XXL (11B).
>
> 2) **Distilled**: Applying KD to T5 models for optimizing downstream task performance. It contains four model points: Distilled-T5-Small (77M), Distilled-T5-Base (250M), Distilled-T5-Large (800M), Distilled-T5-XL (3B). All distilled models use Raw T5-XXL (11B) as the teacher.
>
> 3) **Speculative**: Applying SD to T5 models. It contains four target models and all use Raw T5-Small (77M) as draft model for best performance. The target models are T5-Base (250M), T5-Large (800M), T5-XL (3B), and T5-XXL (11B).
>
> 4) **DistillSpec**: Applying KD to T5 models and using distilled models as target and draft. It has three target model sizes: (1) Distilled-T5-Base (250M) (2) Distilled-T5-Large (800M), and (3) Distilled-T5-XL (3B). We use Distilled-T5-Small (77M) as the draft model for all target models.
>
> > *While DistillSpec is effective on certain tasks, the experiments mainly focus on T5 models and simple tasks (except for GSM8K) ... The effectiveness of DistillSpec should be evaluated using more recent large models and some zero-shot benchmarks to prove its effectiveness.*
>
> Thanks for the suggestion! We have added a new experiment to study the transferability of the distilled draft model across different datasets. Specifically, we take the distilled draft model from GSM8K and evaluate it on all **BigBenchHard** tasks (23 tasks in total) [1] with **zero-shot CoT**. We observe that the distilled draft model transfers well to other datasets. Compared to standard SD, it improves the average decoding speed from 1.93x and 1.78x to 2.21x and 2.02x using the greedy and non-greedy decoding, respectively. See Appendix Figure C.2 and C.3 for a detailed breakdown of the tasks in BBH.
>
> As for the model we use, we choose the T5 model series because they are a set of powerful and representative models, which are carefully studied in the original SD paper [2] and a follow up study in [3]. Further, to demonstrate the effectiveness of our method for the decoder-only model, we followed [2][4]  and had already included an experiment on LM1B in our original submission. As shown in Table C.1, DistillSpec consistently yields speed gain over the vanilla SD. We expect our findings from the LM1B experiment with GPT-like decoder-only models to extend to LLaMA sized models.
>
> [1] Suzgun, M., Scales, N., Schärli, N., Gehrmann, S., Tay, Y., Chung, H. W., ... & Wei, J. (2022). Challenging big-bench tasks and whether chain-of-thought can solve them. arXiv preprint arXiv:2210.09261.
>
> [2] Leviathan, Y., Kalman, M., & Matias, Y. (2023, July). Fast inference from transformers via speculative decoding. In International Conference on Machine Learning (pp. 19274-19286). PMLR.
>
> [3] Kim, S., Mangalam, K., Moon, S., Malik, J., Mahoney, M.W., Gholami, A., & Keutzer, K. (2023). Speculative Decoding with Big Little Decoder.
>
> [4] Sun, Z., Suresh, A. T., Ro, J. H., Beirami, A., Jain, H., & Yu, F. (2023). Spectr: Fast speculative decoding via optimal transport.

---

> > ### Author Response · Authors · 2023-11-19
> > **Author Response: Clarify the generation quality of DistillSpec and provide a decoding example.**
> >
> > > *The paper does not analyze the generated texts by DistillSpec in greater detail, like the diversity and coherence, which are important aspects of text quality and largely influenced by the sampling approaches. It would be helpful to provide some examples of the generated tasks.*
> >
> > Thanks for the suggestion. We agree that text quality can be influenced by the sampling approaches. However, we want to note that the design of the SD algorithm [1][2] ensures that the generated text will be **exactly the same as if it is generated by the target model**. So, if we use a greedy version of SD, the generated text will be exactly the same as the one generated by the target model, while a non-greedy version of SD will guarantee the sampled text follows the same distribution as the target model.
> >
> > As for DistillSpec, it only provides a more aligned draft model without doing any modifications to the target model. **Therefore, SD with a distilled draft won’t change the quality of the generated text and it will be the exact same as the target model.** For illustration purposes, we provide a decoding example from GSM8K using greedy decoding. As you can see, using speculative decoding with a distilled draft or non-distilled draft model does not change the final decoding output. But the distilled draft model reduces the rejected tokens for the entire decoded output from 9 to 3.
> >
> > **Input**: Adrianne is collecting different kinds of beads for making bracelets. Her mother gave her 20 metallic beads. Her sister gave her ten more beads than her mother, and her friend gave her twice as many as her mother gave. How many beads did Adrianne have altogether? Let's think step by step.
> >
> > **T5-XL decode**: Adrianne started with 20 beads from her mother. Her sister gave her 10 more than her mother, so 20 + 10 = 30. Her friend gave her twice as many as her mother, so 30 * 2 = 60. So she had 20 + 30 + 60 = 110 beads.
> >
> > **T5-Small decode**: Adrianne started with 20 beads. Her mother gave her 10 more than her mother, so she gave her 10 + 10 = 22 beads. Her friend gave her twice as many as her mother, so she gave her 2 * 22 = 34 beads. So Adrianne had 20 + 22 + 34 = 70 beads altogether.
> >
> > **Non-distilled draft model speculative decode process.**
> > 1) Adrianne started with 20 beads~~.~~ from
> > 2) Adrianne ... beads from her mother. Her sister gave her 10 more than her mother, so 20 + 10 = ~~40~~ 30
> > 3) Adrianne ... more than her mother, so 20 + 10 = 30. Her friend gave her twice as many as her mother, so ~~2~~ 30
> > 4) Adrianne ... as her mother, so 30 * 2 = 60. So ~~her~~ she
> > 5) Adrianne ..., so 30 * 2 = 60. So she ~~gave~~ had
> > 6) Adrianne ..., so 30 * 2 = 60. So she had ~~60~~ 20
> > 7) Adrianne ..., so 30 * 2 = 60. So she had 20 + ~~60~~ 30
> > 8) Adrianne ..., so 30 * 2 = 60. So she had 20 + 30 + 60 = ~~150~~ 110
> > 9) Adrianne ..., so 30 * 2 = 60. So she had 20 + 30 + 60 = 110 beads ~~altogether~~.
> > 10) **Final Decoded Output**: Adrianne started with 20 beads from her mother. Her sister gave her 10 more than her mother, so 20 + 10 = 30. Her friend gave her twice as many as her mother, so 30 * 2 = 60. So she had 20 + 30 + 60 = 110 beads.
> >
> > **Distilled draft model speculative decode process**
> > 1) Adrianne started with 20 beads from her mother. Her sister gave her 10 more than her mother, so ~~she~~ 20
> > 2) Adrianne started with 20 beads from her mother. Her sister gave her 10 more than her mother, so 20 + 10 = 30 ~~beads~~.
> > 3) Adrianne started with 20 beads from her mother. Her sister gave her 10 more than her mother, so 20 + 10 = 30. Her friend gave her twice as many as her mother, so 30 * 2 = 60. So ~~altogether~~ she
> > 4) **Final Decoded Output**: Adrianne started with 20 beads from her mother. Her sister gave her 10 more than her mother, so 20 + 10 = 30. Her friend gave her twice as many as her mother, so 30 * 2 = 60. So she had 20 + 30 + 60 = 110 beads.
> >
> > [1] Leviathan, Y., Kalman, M., & Matias, Y. (2023, July). Fast inference from transformers via speculative decoding. In International Conference on Machine Learning (pp. 19274-19286). PMLR.
> > [2] Chen, C., Borgeaud, S., Irving, G., Lespiau, J. B., Sifre, L., & Jumper, J. (2023). Accelerating large language model decoding with speculative sampling. arXiv preprint arXiv:2302.01318.

---

> > > ### Author Response · Authors · 2023-11-19
> > > **Author Response: Discussion and comparison to BiLD and Online Speculative Decoding.**
> > >
> > > > *The paper does not compare or discuss DistillSpec with other methods that combine large and small models at inference, especially under the lossy decoding cases.*
> > >
> > > BiLD [1] proposes an SD variant for the case where the decoding speed difference between the target model and draft model is small. In contrast, our paper conducts a comprehensive study of KD for SD and is complementary to BiLD.
> > >
> > > BiLD can be viewed as a SD variant with two additional components 1) fallback policy 2) aligned draft using SeqKD. The rollback policy in BiLD is similar to the rejection sampling of SD, but uses CE between the target and draft as the rejection criterion. Figure C.18 shows that DistillSpec can reduce the CE more effectively than BiLD alignment with SeqKD. The fallback policy reduces the unnecessary computation from the draft model by transferring the decoding to the target model when the draft model is uncertain. This fallback policy will be effective when the draft model computation is expensive, but becomes less effective when the draft model computation is negligible compared to the target model.
> > >
> > > Nevertheless, we conduct some experiments to distill T5-small models from T5-large on XSum and CNNDM. DistillSpec achieves a lossless decoding speed of 1.8x and 2.23x on XSum and CNNDM, respectively. In contrast, aligned BiLD (with SeqKD) only provides a 1.5x and 1.85x speed up for XSum and CNNDM, respectively (we report the numbers from BiLD [1]). As we show in Figure 2 and 3a, SeqKD is not very effective compared to the whitebox KD algorithms that use model logits for distillation. Therefore, DistillSpec can potentially be utilized to improve BiLD, which we leave for future work.
> > >
> > > [1] Kim, S., Mangalam, K., Moon, S., Malik, J., Mahoney, M.W., Gholami, A., & Keutzer, K. (2023). Speculative Decoding with Big Little Decoder.
> > > [2] Leviathan, Y., Kalman, M., & Matias, Y. (2023, July). Fast inference from transformers via speculative decoding. In International Conference on Machine Learning (pp. 19274-19286). PMLR.
> > >
> > > > *I am wondering how DistillSpec will be compared with recent work that combines speculative decoding and distillation [1]. It would be interesting to have this discussion.*
> > >
> > > Online Speculative Decoding is a concurrent work (uploaded after the ICLR submission deadline) that also uses KD to improve the efficiency of speculative decoding. However, their setup is very different from ours where they assume the query distribution is changing over time instead of static, and they are allowed to use the idle FLOPs in the serving system. Besides, their evaluation mainly focuses on the improvement of acceptance rate, without measuring the actual latency speedup. In contrast, our work measures the actual latency speedup besides the acceptance rate improvement.
> > >
> > > [1] Liu, X., Hu, L., Bailis, P.D., Stoica, I., Deng, Z., Cheung, A., & Zhang, H. (2023). Online Speculative Decoding. ArXiv, abs/2310.07177.
> > >
> > > -----
> > > *We hope that most of the reviewer’s concerns have been addressed and, if so, we would appreciate it if they could reconsider their assessment. We’d be happy to engage in further discussions.*

---

> > > > ### Comment · Reviewer_GLMj · 2023-12-02
> > > > **Post-rebuttal**
> > > >
> > > > Dear Authors,
> > > >
> > > > Thank you for the detailed rebuttal and clarifications provided. The additional results and discussions are comprehensive and have mostly addressed my concerns. Therefore, I am willing to increase my score.

---

### Official Review · Reviewer_gZt5 · 2023-11-01

**Soundness:** 3 good
**Presentation:** 2 fair
**Contribution:** 2 fair
**Rating:** 6
**Confidence:** 3

**Summary:**

This paper proposes DistillSpec, which improves the efficiency of speculative decoding (SD) by aligning the distributions (at the token and sequence level) between the draft model and the target model in advance. The paper further provides valuable insights regarding the recipe of distillation data, distillation objective and sampling strategy.

Experiments show that DistillSpec speedup SD by 10-45% while preserving the model performance across four diverse datasets.

**Strengths:**

1. SD is an important direction in accelerating LM inference. The idea of leveraging distillation to speed up SD is novel and very effective.
2. Using acceptance rate as the efficiency measure is well-motivated. Using total variation distance (TVD) objective is simple and straightforward in maximizing this efficiency measure.
3. The authors provide a fast alternative of using student-generated data for training, with a theoretical justification and strong experimental results.

**Weaknesses:**

1. The technical novelty is a bit limited. It is a direct application of existing KD techniques in SD.
2. The performance of the method is task-dependent, posing concerns for using the method in practice. For example, the speedup on WMT En-De is marginal and TVD has varied performance on different tasks.
3. Major experiment results are based on small target models. The results on larger models are not very clear -- in Figure 6, which data points correspond to which sizes of target and draft models?
4. The method is target-model-dependent, meaning that we need to distill a new draft model for each new target model. Such a distillation cost can be quite expensive, especially when using online data generation.

**Questions:**

See weakness.

**Details Of Ethics Concerns:**

No ethics concerns.

---

> ### Author Response · Authors · 2023-11-19
> **Author Response: Clarify the novelty and contributions. Clarify the performance variance of different tasks and divergence functions.**
>
> Thank you for taking the time to provide your thoughtful and valuable feedback on our work. Below are responses to your concerns.
>
> > *The technical novelty is a bit limited. It is a direct application of existing KD techniques in SD.*
>
> We argue that the application of KD in SD is not as straightforward or well-explored. Our approach diverges from the standard KD, which primarily aims at enhancing the student model's performance. Our focus is on improving alignment to enhance decoding efficiency, a distinct objective as evidenced in Figure 5a.
>
> Additionally, our work critically examines the effectiveness of applying KD to SD. Figures 2 and 3a demonstrate that previous attempts, such as the use of SeqKD [1], were inadequate in enhancing SD efficiency. Our research contributes to this field by providing a systematic and comprehensive study (Section 5.2) on the most cost-effective methods of applying KD in SD. The insights we offer, particularly regarding the cost-effectiveness of using student-generated data and the task and strategy-dependent nature of the best divergence function, are valuable for practical applications.
>
> Furthermore, we wish to emphasize our study of the quality and latency trade-off, a central challenge in LLM inference. Our research explores this trade-off through two perspectives: 'lossy speculative decoding' and 'model garden' (Section 5.3). The former offers fine-grained control over the trade-off curve, while the latter presents optimal strategies for combining SD and KD. These contributions are novel and highly pertinent for practitioners seeking to balance quality and latency in model deployment.
>
> [1] Stern, M., Shazeer, N., & Uszkoreit, J. (2018). Blockwise parallel decoding for deep autoregressive models. Advances in Neural Information Processing Systems, 31.
>
>
> > *The performance of the method is task-dependent, posing concerns for using the method in practice. For example, the speedup on WMT En-De is marginal.*
>
> Indeed, the additional speedup provided by DistillSpec is task-dependent. There are two reasons for this:
>
> 1) First, **different tasks have different amounts of headroom**. As shown in Figure 1 and Table D.1, the original SD already shows a speedup of 2.36x on WMT, in contrast to a 1.44 speedup on XSum. It suggests that target model and draft model exhibit different levels of alignment when trained on different tasks and the model trained on WMT is already much more aligned than XSum. Thus, the headroom for improvement is smaller for WMT than XSum. The difference in task nature was also observed in previous SD literature [1][2].
>
> 2) Secondly, **we did not attempt to optimize the hyper-parameters for each task**: we use the same hyperparameters for all distillation experiments, so they may be suboptimal for each task. Figure C.8 and C.9 show the progression of the acceptance rate tested on the sequence generated by the target model. The unsaturated learning curves suggest that we can achieve much better performance if we train the model longer, especially in the WMT case. However, our current training strategy can already show the effectiveness of the DistillSpec and training longer to optimize the absolute performance gain might not provide any new insights.
>
> [1] Leviathan, Y., Kalman, M., & Matias, Y. (2023, July). Fast inference from transformers via speculative decoding. In International Conference on Machine Learning (pp. 19274-19286). PMLR.
>
> [2] Chen, C., Borgeaud, S., Irving, G., Lespiau, J. B., Sifre, L., & Jumper, J. (2023). Accelerating large language model decoding with speculative sampling. arXiv preprint arXiv:2302.01318.
>
>
> > *TVD has varied performance on different tasks.*
>
> **We argue that “TVD has varied performance on different tasks” is itself a finding of our paper.** Although TVD is the theoretically motivated objective we should optimize, our systematic study in Section 5.2 suggests that it does not consistently yield the best empirical performance. While this gap between theory and practice is disappointing, we think it truthfully reflects the fact that distillation in the real world is challenging.
>
> Just like ML algorithms that have some hyperparameters to tune, we think the divergence function is one of the hyperparameters in a distillation algorithm and you should tune it for your task. Note that this process only needs to be done once and can be amortized across many serving queries.
>
> Overall, we view treating the divergence function as a hyperparameter as an interesting take-away from our study.

---

> ### Author Response · Authors · 2023-11-19
> **Author Response: Clarify the model sizes and distillation cost. Add new experiments on the transferability of the distilled draft models (different target sizes, GSM8K ->BBH).**
>
> > *Major experiment results are based on small target models.*
>
> Our main results (Figure 1) are based on the second large T5 model (T5-XL, 3B) as target and T5 (T5-small, 77M) as draft. Besides, we include additional results using the largest T5 (T5-XXL, 11B) as target in Appendix (Figure C.1). Compared to recent literature on SD, where BiLD [1] only considers T5-Large (770M) as the target model, SpecTr [2] considers a 97M decoder only model, Spector & Re [3] consider a 762M GPT2 model, our target model choices (3B, 11B) are not small, which closely follows the original SD paper [4].
>
> [1] Kim, S., Mangalam, K., Moon, S., Malik, J., Mahoney, M.W., Gholami, A., & Keutzer, K. (2023). Speculative Decoding with Big Little Decoder.
>
> [2] Sun, Z., Suresh, A. T., Ro, J. H., Beirami, A., Jain, H., & Yu, F. (2023). Spectr: Fast speculative decoding via optimal transport.
>
> [3] Spector, B., & Re, C. (2023). Accelerating LLM inference with staged speculative decoding. arXiv preprint arXiv:2308.04623.
>
> [4] Leviathan, Y., Kalman, M., & Matias, Y. (2023, July). Fast inference from transformers via speculative decoding. In International Conference on Machine Learning (pp. 19274-19286). PMLR.
>
> > *The results on larger models are not very clear -- in Figure 6, which data points correspond to which sizes of target and draft models?*
>
> We agree and have updated the figure. The target and draft model size are detailed below.
>
> 1) **Raw**: Deploying supervised fine-tuned (SFT) T5 models. It contains five model points: T5-Small (77M), T5-Base (250M), T5-Large (800M), T5-XL (3B), and T5-XXL (11B).
>
> 2) **Distilled**: Applying KD to T5 models for optimizing downstream task performance. It contains four model points: Distilled-T5-Small (77M), Distilled-T5-Base (250M), Distilled-T5-Large (800M), Distilled-T5-XL (3B). All distilled models use Raw T5-XXL (11B) as the teacher.
>
> 3) **Speculative**: Applying SD to T5 models. It contains four target models and all use Raw T5-Small (77M) as draft model for best performance. The target models are T5-Base (250M), T5-Large (800M), T5-XL (3B), and T5-XXL (11B).
>
> 4) **DistillSpec**: Applying KD to T5 models and using distilled models as target and draft. It has three target model sizes: (1) Distilled-T5-Base (250M) (2) Distilled-T5-Large (800M), and (3) Distilled-T5-XL (3B). We use Distilled-T5-Small (77M) as the draft model for all target models.
>
> > *The method is target-model-dependent, meaning that we need to distill a new draft model for each new target model. Such a distillation cost can be quite expensive, especially when using online data generation.*
>
> SD’s efficiency depends on the alignment between the target model and draft model, which is “model dependent”. When we have a new target model to deploy, if it is very different from the previous target model, then, it is reasonable to distill a new draft model tailored to the new target model. However, if the new target model is very similar to the old target model (e.g., they belong to the same model class, but are of different sizes), then it is sensible to reuse the existing distilled draft model. As we demonstrated in Appendix Figure C.1, the distilled draft model from T5-XL (3B) as target also works well for T5-XXL (11B) as target. Though it is a suboptimal choice, it still achieves a consistent improvement of 7%- 37% over original SD across four different datasets.
>
> In contrast, if the new target model is very different, then the extra distillation cost is inevitable to realize significant gains via SD. Note that the distillation process only needs to be done once, which can be amortized across many serving queries and the real cost is serving to a large number of users. As for the online data generation, Figure 3a suggests that using the student model for online data generation is actually the most cost-effective way to do distillation. In contrast, when you use a static ground truth dataset (i.e., SupKD), your model performance will soon plateau.
>
> **Additionally, we have added a new experiment to study the transferability of the distilled draft model across different datasets.** Specifically, we take the distilled draft model from GSM8K and evaluate it on all BigBenchHard tasks (23 tasks in total) [1] with zero-shot CoT. We observe that the distilled draft model transfers well to other datasets. Compared to standard SD, it improves the average decoding speed from 1.93x and 1.78x to 2.21x and 2.02x using the greedy and non-greedy decoding, respectively. See Appendix Figure C.2 and C.3 for a detailed breakdown of the tasks in BBH.
>
> [1] Suzgun, M., Scales, N., Schärli, N., Gehrmann, S., Tay, Y., Chung, H. W., ... & Wei, J. (2022). Challenging big-bench tasks and whether chain-of-thought can solve them. arXiv preprint arXiv:2210.09261.
>
> -----
> *We hope that your concerns/questions have been addressed and, if so, we would appreciate it if you could reconsider your assessment. We’d be happy to engage in further discussions.*

---

> > ### Comment · Reviewer_gZt5 · 2023-12-04
> >
> > Thanks for providing clarifications for the experiments and adding results on transferability. The response has cleared my concerns and I would like to keep the score.

---

### Author Response · Authors · 2023-11-19
**General Response and Updates in Revision**

We express our sincere gratitude to all the reviewers for their valuable time and effort during the review process. All reviewers agree that our work provides novel insights on Distillation and Speculative Decoding and find our experiments “strong”, “comprehensive and systematic” and “quite solid”. We have addressed each reviewer's comments individually and revised the paper accordingly.

Our major changes to the paper are highlighted in purple text and listed below:

1. **[Experiment, @gZt5, @GLMj, @p4vK]** Transferability Evaluation: evaluate distilled draft model trained on GSM8K on 23 BigBenchHard tasks using zero-shot CoT, which results in average speedups of 28% and 23% using greedy and non-greedy decoding. (Figure 1, Figure C.1-C.5)
2. **[Clarification, @gZt5, @GLMj, @p4vK]** We update Figure 6 to clarify the confusion regarding model sizes.
3. **[Clarification & Related Work, @GLMj]** We discuss and compare BiLD [1] on XSum / CNNDM and discuss the concurrent work on Online Speculative Decoding [2]. We add the definition of DistillSpec to the main text and clarify how we generate data from models. We also add more experimental details in Appendix A.1.

[1] Kim, S., Mangalam, K., Moon, S., Malik, J., Mahoney, M.W., Gholami, A., & Keutzer, K. (2023). Speculative Decoding with Big Little Decoder.

[2] Liu, X., Hu, L., Bailis, P.D., Stoica, I., Deng, Z., Cheung, A., & Zhang, H. (2023). Online Speculative Decoding. ArXiv, abs/2310.07177.

---

### Author Response · Authors · 2023-11-22
**Discussion period nears its end, please let us know if you have any remaining questions after reading our response.**

Dear Reviewers,

Thank you again for taking the time to review our submission and providing your insightful feedback. In our response, we have tried our best to address your comments/concerns/questions. As the author-reviewer discussion period nears its end, please let us know if you have any remaining questions after reading our response.

---

### Meta-Review · Area_Chair_jKBf · 2023-12-11

**Metareview:**

This paper presents an idea to improve speculative decoding using distillation to better align the draft model as well as the base model.

Strength:
1. An important topic with strong empirical results.

Weakness:
1. Limited novelty by applying distillation to the draft model. The authors did provide some useful lessons for practitioners through studying trade offs for different methods.

**Justification For Why Not Higher Score:**

Limited novelty.

**Justification For Why Not Lower Score:**

This paper has limited novelty. However given it is a very important topic, the findings of this paper could still be potentially useful for many practitioners in this area.

---

### Decision · Program_Chairs · 2024-01-16

Accept (poster)